# Cross-linking mass spectrometry uncovers protein interactions and functional assemblies in synaptic vesicle membranes

Sabine Wittig[1], Marcelo Ganzella[2], Marie Barth [1], Susann Kostmann[1], Dietmar Riedel [2], Ángel Pérez-Lara [2,3], Reinhard Jahn[2] & Carla Schmidt [1✉]

Synaptic vesicles are storage organelles for neurotransmitters. They pass through a trafficking cycle and fuse with the pre-synaptic membrane when an action potential arrives at the nerve terminal. While molecular components and biophysical parameters of synaptic vesicles have been determined, our knowledge on the protein interactions in their membranes is limited. Here, we apply cross-linking mass spectrometry to study interactions of synaptic vesicle proteins in an unbiased approach without the need for specific antibodies or detergent-solubilisation. Our large-scale analysis delivers a protein network of vesicle sub-populations and functional assemblies including an active and an inactive conformation of the vesicular ATPase complex as well as non-conventional arrangements of the luminal loops of SV2A, Synaptophysin and structurally related proteins. Based on this network, we specifically target Synaptobrevin-2, which connects with many proteins, in different approaches. Our results allow distinction of interactions caused by 'crowding' in the vesicle membrane from stable interaction modules.

[1] Interdisciplinary Research Centre HALOmem, Charles Tanford Protein Centre, Institute for Biochemistry and Biotechnology, Martin Luther University Halle-Wittenberg, Halle, Germany. [2] Department for Neurobiology, Max Planck Institute for Biophysical Chemistry, Göttingen, Germany. [3]Present address: Department of Physical Chemistry, Faculty of Pharmacy, University of Granada, Granada, Spain. ✉email: carla.schmidt@biochemtech.uni-halle.de

Signal transmission between neurons is mediated by exocytosis of neurotransmitters and takes place at specifically designed contact sites called synapses[1]. In the pre-synaptic nerve terminal, neurotransmitters are stored in synaptic vesicles. They are released from the vesicles into the synaptic cleft where they are received by neurotransmitter receptors of the post-synaptic membrane. To allow rapid and directed neurotransmitter release, synaptic vesicles undergo a trafficking cycle which prepares them for repeated rounds of exocytosis[2]. The synaptic vesicle cycle starts with neurotransmitter uptake operated by neurotransmitter-specific transporters. These transporters are fuelled by the proton-electrochemical gradient established by a vesicular proton pump with ATPase activity (termed 'V-ATPase')[3,4]. After neurotransmitter loading, synaptic vesicles form a readily releasable pool of docked vesicles at the active zone of the pre-synaptic terminal[5]. Docked vesicles are then activated in a process called 'priming'[6]. Upon intracellular $Ca^{2+}$ influx as a response of an action potential, the vesicle and pre-synaptic membranes fuse and neurotransmitters are released[7,8]. Synaptic vesicles are retrieved from the plasma membrane following alternative pathways, including clathrin-mediated endocytosis or transient pore formation and subsequent pore closure ('kiss-and-run')[9].

First and foremost, synaptic vesicles are storage organelles for neurotransmitters. However, in addition to the neurotransmitter loading machinery, they are densely packed with proteins that orchestrate the many interactions formed during the synaptic vesicle cycle and fulfil the required functional tasks. Membrane fusion, for instance, is realised by SNARE (i.e. soluble N-ethyl-maleimide-sensitive-factor attachment receptor) complex formation including interactions between Syntaxin-1A and SNAP25 on the pre-synaptic membrane and Synaptobrevin-2 on the vesicular membrane[10]. For this, Syntaxin-1A and Synaptobrevin-2 both contribute one and SNAP25 two α-helices to the stable four-helix bundle of the assembled SNARE complex[11]. In addition to the core fusion machinery, there are several regulatory factors which are anchored in the respective membranes or are components of the synaptic cytosol[12]. An example is the vesicular calcium sensor Synaptotagmin-1, which binds negatively charged phospholipids as well as the SNARE complex suggesting different regulatory modes of action[13].

To unravel the complexity of synaptic vesicles, several studies targeted the proteomes of purified synaptic vesicles[14–16] as well as specific states such as docked[17] or clathrin-coated vesicles[18] or GABAergic and glutamergic sub-populations[19]. These studies revealed a set of synaptic vesicle core components; however, the functional role of many vesicle proteins remains elusive. Quantitative studies then delivered copy numbers of vesicular proteins and provided a first glimpse into the organisation of the crowded vesicle membrane[15,20] and its synaptic environment[21]. Accordingly, Synaptobrevin-2 represents the major protein component with up to 70 copies per vesicle. Additional abundant vesicle components are Synaptophysin, Synaptotagmin-1 and Synapsin-1. In contrast, only one or two copies of the V-ATPase complex are anticipated in synaptic vesicles[15,20].

Despite the progress in unravelling the synaptic vesicle proteome, we have only little knowledge on the protein interactions formed between synaptic vesicle components at the different stages of the synaptic vesicle cycle. Apart from the SNARE complex, for which high-resolution structures are available[11,22], our knowledge mostly relies on binary or ternary interactions identified in biochemical assays. A frequent target of such studies were the two vesicle markers Synaptobrevin-2 and Synaptophysin. Chemical cross-linking and subsequent immunoblotting revealed oligomerisation of Synaptophysin[23–25] as well as a stable complex between the two proteins[26]. These interactions were further investigated by electron microscopy suggesting a 1:2 (Synaptophysin:Synaptobrevin-2) stoichiometry in a hexameric complex[27,28]. Immunoprecipitations from synaptosome extracts further revealed interactions between Synaptophysin/Synaptobrevin-2 and the membrane-sector $V_O$ of the V-ATPase[29] as well as several detergent-sensitive interactions between SV2A, Synaptotagmin-1, Synaptophysin and Synaptobrevin-2[30]. Recent studies targeted intact synaptosomes and provided a three-dimensional model of an 'average' synapse[21] as well as protein interaction networks[31]. However, a detailed study on purified synaptic vesicles is still missing. One possible reason is the heterogeneity of synaptic vesicle populations making them challenging targets for structural biology techniques.

Here, we overcome these difficulties by combining chemical cross-linking and mass spectrometry (MS) with biochemical and biophysical tools to unravel the protein interactions of purified synaptic vesicles. In a first set of experiments, we present an interaction network of synaptic vesicle proteins revealing different co-existing functional assemblies. We specifically target the V-ATPase complex and identify two conformations, the active, fully assembled enzyme and the dissociated, inhibitory $V_O$-ATPase. We then follow different approaches to unravel the interactions of co-existing sub-networks and provide insights into specific and stable interaction modules. In particular, the fusion of synaptic vesicles with 'empty' liposomes reveals specific protein interactions formed in a spacious membrane environment allowing their distinction from unspecific interactions formed in the naturally crowded vesicle membrane.

## Results

**Proteome assessment of synaptic vesicles purified from rat brain.** We set out to study protein interactions in synaptic vesicles using chemical cross-linking and MS. To allow unambiguous assignment of proteins in these large-scale experiments, we first evaluated five independent preparations of synaptic vesicles. For this, synaptic vesicles were purified from rat brain following established protocols[15,32,33]. Figure 1a shows a gel of a typical vesicle preparation. Negative stain electron microscopy further confirmed integrity of the vesicles (Fig. 1b and Supplementary Fig. 1). Proteins separated by gel electrophoresis were hydrolysed using endoproteinase trypsin and obtained peptides were analysed by liquid chromatography-coupled tandem-MS (LC-MS/MS). Gel electrophoresis and LC-MS/MS analysis confirmed that different vesicle preparations are comparable (Supplementary Fig. 2). In total, when applying a false-discovery rate (FDR) of 1%, we identified >1000 proteins with high scores (Supplementary Data 1 and Methods) including all previously identified synaptic vesicle proteins[14–16] as well as typical contaminants. We further used intensity-based absolute quantification (iBAQ)[34] to evaluate protein abundances. For this, the sum of peptide intensities corresponding to one protein is normalised by the number of theoretically observable peptides. The relative iBAQ then represents the relative proportion of each protein in the sample and allows the estimation of protein stoichiometries. As expected, all major synaptic vesicle components are amongst the high abundant proteins (Supplementary Data 1).

We first inspected the V-ATPase complex which represents an individual multi-subunit protein complex embedded in the vesicle membrane and represented on its cytosolic side. A recently published high-resolution structure from rat brain revealed the exact composition and arrangements of protein subunits in the soluble $V_1$- and the membrane-embedded $V_O$-domains[35]. Our proteomics analysis confirmed the presence of 15 of the 16 reported subunits including isoforms of subunits B, E, G, C, a and d (Supplementary Data 2). Subunit e/e2 does not contain a sufficient number of tryptic cleavage sites and is therefore missing in our analysis. Subunits (isoforms) B1, C2, a4, c'', d2, f and Ac45 were only identified with low scores and therefore failed our stringent

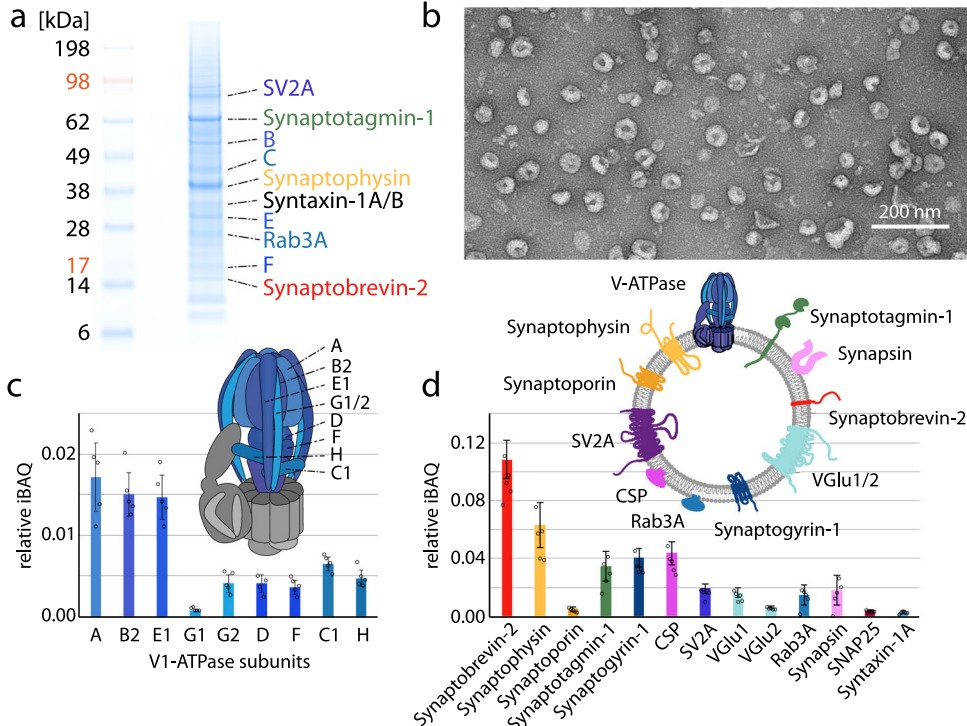

**Fig. 1 Proteomic characterisation of synaptic vesicles. a** SDS-PAGE of a typical preparation of synaptic vesicles (*n* > 5). Selected major protein components are indicated. **b** Electron microscopy image of purified synaptic vesicles confirming integrity of the vesicles (*n* = 3). **c** Relative iBAQ (intensity-based absolute quantification) of the $V_1$-ATPase. Individual data points, the mean and the standard deviation from five biological replicates are given. **d** Relative iBAQ of selected synaptic vesicle proteins. Individual data points, the mean and the standard deviation from five biological replicates are given. The topology of major vesicle proteins is shown.

selection criteria (see 'Methods'). Subunit c" was to-date only confirmed in the recent high-resolution structure[35]. Due to a low number of tryptic cleavage sites, we identified only one tryptic peptide in one of five preparations (Supplementary Data 2). Importantly, subunit H, which is required for full activity of the enzyme[36] but was missing in the recent structure[35], was identified in our analyses with high scores and sequence coverage (Supplementary Data 1 and 2) confirming the quality of our vesicle preparations.

Having confirmed the presence of the intact V-ATPase, we used the ratio of iBAQ intensities for assessing protein stoichiometries. With one exception (subunit G1/G2), the expected stoichiometry of $A_3B_3E_3G_3D_1F_1C_1H_1$ was observed for the $V_1$-domain (Fig. 1c). Stator subunit G1/G2 was under-represented when compared with its binding partner E1 and core subunits A and B2. Close inspection of the amino acid sequence revealed an atypical distribution of tryptic cleavage sites likely affecting iBAQ analysis which is based on the number of observable peptides. Due to low identification scores of some $V_O$ subunits (see above), only subunits c1, a1 and d1 were considered for stoichiometry estimation. In agreement with the high-resolution structure[35], subunits a1 and d1 show equal abundances and subunit c1 is present at higher quantities (Supplementary Data 1). The expected 9-fold abundance of subunit c1 was, however, not observed. Most likely, similar reasons as described above for subunit G1/G2 cause these discrepancies for membrane proteins. Comparing iBAQ values of $V_1$ and $V_O$ domains reveals a higher abundance of $V_O$-subunits suggesting that some V-ATPase complexes dissociated as reported previously[37–39].

Our observations made for the V-ATPase agree well with previous studies and the recently published structure. We therefore conclude that stoichiometry estimations can also be made for other synaptic vesicle proteins. In agreement with previous studies, we found that Synaptobrevin-2 is the most abundant protein in synaptic vesicles[15,21]. Relative abundances of the other major protein components correlate well with previously established copy numbers (Supplementary Data 1). To name a few examples, Synaptobrevin-2, Synaptophysin and Synaptotagmin-1 displayed relative abundances of approximately 4:2:1 correlating well with previously published copy numbers of 70 (Synaptobrevin-2) versus 31 (Synaptophysin) versus 15 (Synaptotagmin-1)[15]. Proteins that showed comparable amounts in previous studies were also equal-abundant in our analyses; examples are Synaptogyrin and CSP (approximately 2 copies[15]), SV2A and VGlu (approximately 15 copies[21]) as well as Synapsin and Rab3A (approximately 10 copies[15]) (Fig. 1d). Importantly, Syntaxin-1A and SNAP25, which are constituents of the SNARE complex and infiltrate into the vesicle membrane from the presynaptic membrane during endocytosis, are only low abundant and have similar iBAQ values suggesting that they enter the vesicle membrane together in the form of residual or pre-formed SNARE complex.

**Cross-linking reveals inhibited and functionally active states of the V-ATPase complex in synaptic vesicles**. Having identified the proteins present in typical synaptic vesicle preparations, we next studied their protein interactions formed in the vesicle membrane. For this, purified synaptic vesicles were incubated with Bis(sulfosuccinimidyl) suberate (BS3) cross-linker, which covalently links primary amines of lysine side chains and protein N-termini. Following standard protocols[40,41], cross-linked proteins were hydrolysed with trypsin and low abundant, cross-linked peptide pairs were enriched by size-exclusion chromatography. Fractions containing cross-linked peptides were analysed by LC-MS/MS followed by database searching and manual

validation of the mass spectra as suggested previously[42]. To allow large-scale interactomics and, at the same time, compensate for computational space during database search, we generated a synaptic vesicle database containing the top 400 proteins of our proteome analysis (Supplementary Data 1). Applying this strategy, we validated 175 inter- and 297 intra-molecular cross-links in four biological replicates (Supplementary Data 3 and Supplementary Fig. 3). Of these, 80 inter- and 199 intra-molecular cross-links were identified in at least two replicates (Supplementary Fig. 3). 19 of the inter-molecular cross-links originate from homo-oligomers which are identified by equal or overlapping peptide sequences. Example spectra of cross-linked peptides are shown in Supplementary Fig. 4.

Again, we first reviewed cross-links in the V-ATPase complex. We identified and validated 22 inter- and 57 intra-molecular cross-links in ten of the 16 subunits of the V-ATPase including isoforms G1 and G2. We visualised these cross-links in a network plot and compared the inter- and intra-molecular interactions with the high-resolution structure of the V-ATPase complex from rat brain obtained recently[35]. All inter-molecular interactions are observed between protein subunits in close proximity. Specifically, pronounced cross-linking was observed between the 'head' of the V-ATPase (i.e. subunits A and B2) and subunits E1 or D which contribute to the peripheral and central stalks[43]. Interactions between peripheral and central stalk subunits E1 and G2 as well as D and F were also observed. In addition, many cross-links were identified between the N-terminal, soluble part of membrane-bound subunit a1 and subunits d1 or G1/2. Intra-molecular cross-links were mostly observed within subunits A, B2, E1 and C1 highlighting their accessibility for the BS3 cross-linker as well as in the N-terminal part of subunit a1 confirming a high flexibility as suggested previously[35,44,45] (Fig. 2a). For validation of our cross-linking approach, we mapped the observed cross-links into the available high-resolution structure of the V-ATPase complex (Fig. 2b). 74 out of 79 cross-links could successfully be mapped; the missing cross-links were observed in structural regions which are not included in the high-resolution structure or subunits missing in the atomic model. The majority of cross-links (i.e. 50 out of 74) satisfies a cross-linking distance of ≤30 Å (Supplementary Data 4) as expected for the BS3 cross-linker[46]. Importantly, long-distance cross-links (>30 Å) are located in flexible subunits such as the peripheral stalks or subunit a1 (Supplementary Data 4)[47,48]. Assuming random cross-linking of identified cross-linked lysine residues shows a broader distribution with longer cross-linking distances (Supplementary Fig. 5) further confirming our approach and validation.

V-ATPase activity is mainly regulated by reversible dissociation of $V_1$- and $V_O$-domains[37–39]. Close inspection of the interactions obtained for subunit a1 reveals cross-links which exclusively match one of two functional states: an open conformation representing the active, fully assembled enzyme (Fig. 2c) or a closed conformation locking the $V_O$-domain (Fig. 2d). In the open, non-inhibitory conformation, the N-terminal part of subunit a1 makes contact with the peripheral stalk subunit G1/G2 thereby allowing enzymatic activity (cross-links $G1^{K21}$-$a1^{K50}$ and $G2^{K21}$-$a1^{K70}$). In this conformation, interactions between subunits a1 and d1 are not observed (Fig. 2c). In the closed, inhibitory conformation, $V_1$ dissociated from the enzyme and the N-terminal part of a1 folds over the membrane ring. Importantly, interactions observed for this conformation are formed between the N-terminus of a1 and subunit d1 as well as the C-terminal, cytosolic peptide of PRR (residues 324–350) which are both only accessible when the V1-domain dissociates. Cytosolic PRR contains only one accessible lysine residue which was found to cross-link with both a1 ($PRR^{K346}$-$a1^{K70}$) and d1 subunits ($PRR^{K346}$-$d1^{K239}$) (Fig. 2d). These cross-links confirm our observation from the proteomic analysis that the V-ATPase partly dissociates in synaptic vesicles and the $V_O$-domain is over-represented as shown by comparably high iBAQ intensities for $V_O$ subunits (see above and Supplementary Data 1).

## Interaction networks in synaptic vesicles suggest a central role for Synaptobrevin-2

We next inspected cross-links obtained for all synaptic vesicle proteins. For this, inter-molecular cross-links were visualised in a network plot (Fig. 3). The resulting interaction network shows 56 proteins of which 33 are considered synaptic vesicle components[15]. Importantly, most interactions observed in only one biological replicate include contaminants or low abundant proteins. Interactions identified in two or more replicates constitute a network of 76 inter-molecular cross-links including interactions between Synapsin-1, Synaptogyrin-1, Synaptobrevin-2, SV2A/B, Rab3A, CSP, Synaptophysin, Synaptoporin and the AP2 and V-ATPase complexes (Fig. 3). Strikingly, Synaptobrevin-2 was found cross-linked with 32 of the 56 proteins. This includes interactions with specific vesicle proteins as well as low abundant contaminants suggesting a central role for Synaptobrevin-2 in complex formation in synaptic vesicle membranes. Interactions between Synaptobrevin-2 and Synaptotagmin-1, Synaptogyrin-1 or the vesicular transporters (VGlu1/2) are only of low count. In addition to interactions with individual protein subunits, we also identified cross-links between Synaptobrevin-2 and subunits of the AP2 complex. The AP2 complex associates with synaptic vesicles during clathrin-mediated endocytosis and retrieval of Synaptobrevin-2 through different motifs has been described[9]. We therefore assume that Synaptobrevin-2 is a cargo of the AP2 complex and a small population of endocytotic synaptic vesicles was co-purified.

In addition to the dominant, Synaptobrevin-2-centered network, we observed four smaller, ternary protein networks between Synaptobrevin-2, Synaptophysin and (i) V-ATPase-a1, (ii) SV2A, (iii) Synapsin-1 or (iv) CSP (Fig. 3). Importantly, all these ternary networks include Synaptobrevin-2 and Synaptophysin suggesting a predominant interaction of these two proteins as reported previously[26,29]. Multimers (dimers) of Synaptobrevin-2, Synaptophysin and CSP were also observed. These were unambiguously identified by overlapping or identical peptide sequences in the mass spectra of cross-linked peptides (see Supplementary Fig. 4C for an example). To define the oligomeric states of Synaptobrevin-2 and Synaptophysin, we visualised the multimers by immunostaining with specific antibodies after western blotting of cross-linked synaptic vesicle proteins. Following this strategy, multimers up to pentamers were clearly observed for Synaptobrevin-2 while dimers were observed for Synaptophysin (Supplementary Fig. 6). In addition, both antibodies stained one additional protein band corresponding in mass to the cross-linking product of Synaptobrevin-2 and Synaptophysin further confirming this specific interaction.

## Synaptophysin, Synaptobrevin-2 and SV2A interact through luminal loops

Close inspection of the observed protein interactions formed in our network reveals many interactions that are accomplished through luminal loops. This mainly concerns cross-links between SV2A and Synaptophysin but also interactions between Synaptophysin and Synaptogyrin-1 or Synaptophysin and Synaptoporin as well as interactions that mediate Synaptophysin dimerisation (Fig. 4). In few cases, interactions between cytosolic and luminal loops were observed (for instance between Synaptobrevin-2 and SV2A). The observation that these luminal loops are involved in protein interactions in synaptic vesicles when using a non-membrane permeable cross-linker such as BS3 raises the questions whether integrity of the vesicles and, as a

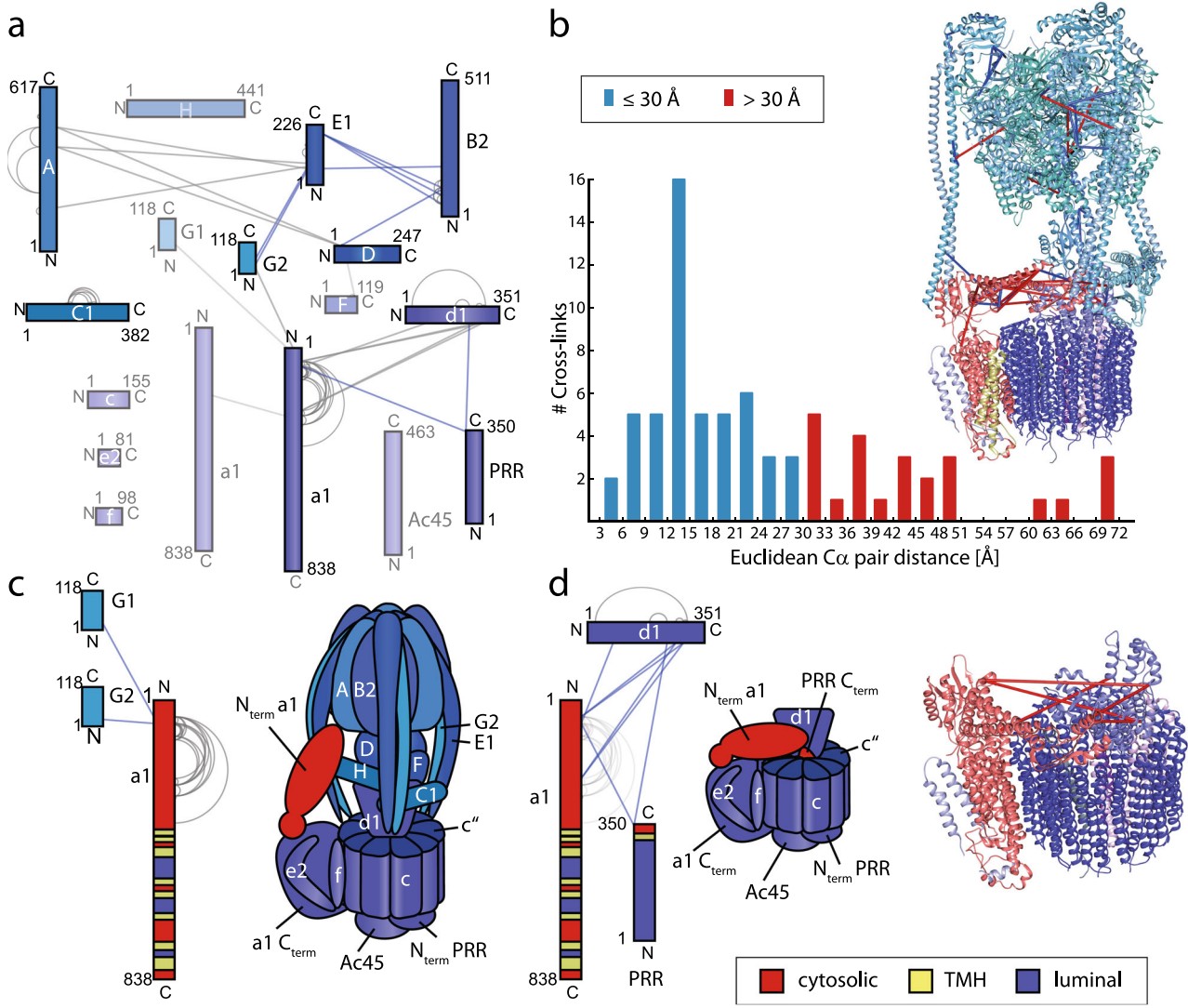

**Fig. 2 Protein interactions in the V-ATPase complex. a** Proteins are shown as coloured bars. The length of the bars corresponds to the protein length. Cross-links identified within (grey) or between (blue and grey) ATPase subunits are shown in an interaction network. Inter-molecular cross-links identified in one (grey) or at least two (blue) biological replicates are shown. **b** 74 out of 78 observed cross-links could be mapped onto the available high-resolution structure of the V-ATPase complex from rat brain (PDB ID 6VQ6). Cross-linking distances and the number of cross-links obtained from at least one biological replicate that satisfy these distances are plotted in a histogram. 50 cross-linked amino acid residues showed distances <30 Å (blue). Long-distance cross-links (red) were mostly observed in flexible protein subunits. Note that cross-links of subunits that are present in multiple copies are only mapped once. **c** Inter-molecular cross-links between subunit a1 and subunit G1/G2 supporting the active conformation are shown. The N-terminal domain of a1 is highlighted in red. A structural cartoon of the V-ATPase is shown for comparison. **d** Cross-links between subunit a1 and subunits d1 and PRR supporting the closed, inhibitory conformation are shown. The N-terminal domain of a1 as well as the C-terminal, cytosolic peptide of PRR are highlighted in red. A structural cartoon of this conformation is shown. A section of the high-resolution structure showing long-distance cross-links supporting the closed conformation is shown. Note that PRR$^{K346}$ is missing in the high-resolution structure and cross-links between V-ATPase-a1 and PRR cannot be shown. Legend to panels (**c**) and (**d**): cytosolic, cytosolic part; TMH, transmembrane helix; luminal, luminal part.

consequence, correct protein orientation are affected, or whether these loops integrate into the vesicle membrane and are therefore accessible for protein interactions.

To rule out the possibility that vesicles lost integrity during the purification process (see also Fig. 1b and Supplementary Fig. 1 for comparison) and to prove that protein orientation is correct, we made use of Botulinium neurotoxin B (BoNT B) which cleaves Synaptobrevin-2, the major protein component of synaptic vesicles, between residues Q76 and F77. The BoNT B cleavage product can then be identified by a mass shift of the protein band during western blotting. For this, we titrated the cell supernatant of a BoNT B producing clostridium bacterial species to aliquots of purified synaptic vesicles and monitored cleavage of Synaptobrevin-2 by

western blotting using a specific antibody (Supplementary Fig. 7). Even at low amounts of BoNT B, Synaptobrevin-2 was completely cleaved without remaining intact Synaptobrevin-2 confirming that the orientation of proteins in synaptic vesicles was not compromised and that vesicle integrity was maintained.

To address the question whether the luminal loops are indeed accessible from the cytosol, we then followed a labelling strategy introduced previously to assess solvent accessible amino acid residues of proteins and protein complexes[49]. Using Sulfo-N-hydroxysuccinimide-acetate (Sulfo-NHS-Acetate) and diethylpyrocarbonate (DEPC), we labelled accessible lysine, histidine, serine, threonine, tyrosine and cysteine residues[50,51] of synaptic vesicle proteins. Both labelling reagents are non-membrane permeable

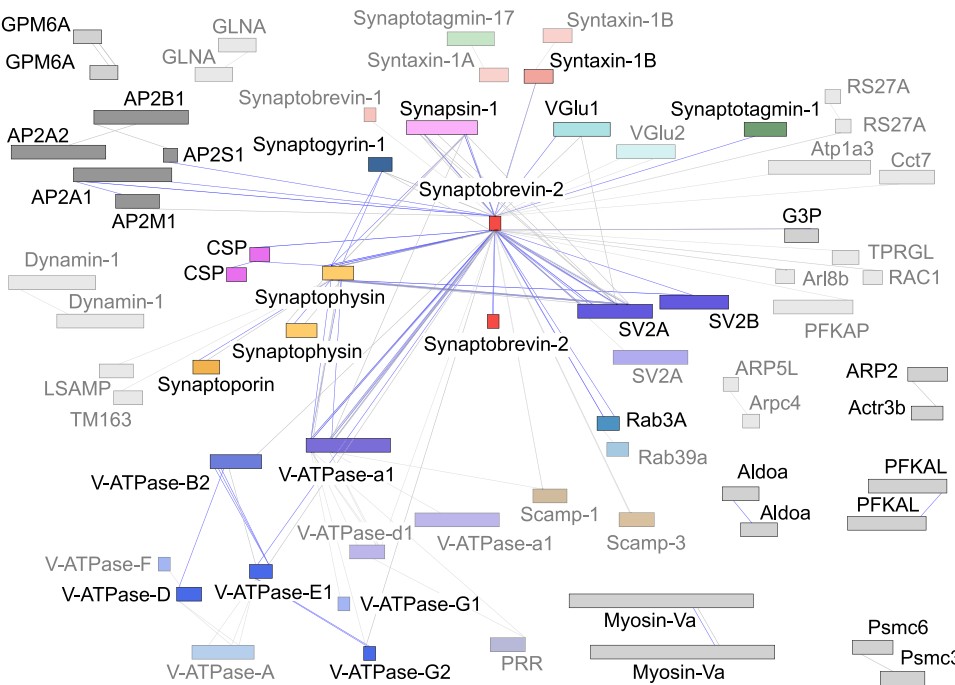

**Fig. 3 Protein interaction network of unstimulated, untreated synaptic vesicles.** Synaptic vesicle proteins are shown as coloured bars; contaminants are shown as grey bars. The length of the bars corresponds to the protein length. The N-terminus is on the left and the C-terminus is on the right side of the bar. Inter-molecular cross-links identified in one (grey) or at least two (blue) biological replicates are shown. Proteins that are linked through interactions from only one biological replicate are transparent.

and only modify amino acid residues that are solvent accessible. Indeed, we found that, in addition to amino acids located in cytosolic domains, amino acids of the large and luminal loops of SV2A, Synaptophysin and Synaptoporin are highly modified (Fig. 4 and Supplementary Data 5). Only one luminal amino acid (K136) was modified in Synaptogyrin-1, however, the number of reactive amino acids and tryptic cleavage site is comparably low in this loop and does not allow identification of more sites. As a control, we inspected Synaptotagmin-1, which contains a large cytosolic part (residues 80–421) and a shorter luminal terminus (residues 1–57), as well as PRR, which contains a short cytosolic peptide (residues 324–350) and a large luminal domain (residues 18–302). Both proteins contain several potential labelling sites in their cytosolic and luminal parts. As expected, only cytosolic amino acids of Synaptotagmin-1 were modified (Supplementary Data 5). In addition, we observed one DEPC-labelled residue (K346) in the cytosolic peptide of PRR. Note that the cytosolic peptide of PRR only contains one tryptic cleavage site and this is the only residue that can be identified. Modified residues were not identified in the luminal regions of neither Synatotagmin-1 nor PRR. Interestingly, by identifying multiple labelled residues in subunits d1 and a1, our labelling approach further confirms the presence of the two V-ATPase conformations as observed in our cross-linking experiments. We therefore conclude that the labelling approach followed here is applicable to identify accessible amino acid residues. Accordingly, luminal loops of SV2A, Synaptophysin, Synaptoporin and Synaptogyrin are accessible in synaptic vesicle membranes and contribute to formation of protein interactions.

In detail, amino acids of the large cytosolic loops of SV2A (residues 1–169 and 356–447) are multiply labelled and cross-link with intra-molecular lysine residues or lysine residues of Synaptobrevin-2 and Synaptophysin (Fig. 4a, b). Many of these interactions are formed between cytosolic and luminal domains (e.g. SV2A$^{K385}$-Synaptophysin$^{K83}$). The large luminal loop of

SV2A (residues 469–598) mainly interacts with the two luminal loops of Synaptophysin (residues 44–101 and 157–194). Synaptobrevin-2 also interacts with Synaptophysin; interestingly, cross-links indicative for these interactions were mostly identified between the cytosolic lysine residues of Synaptobrein-2 and the luminal lysine residues of Synaptophysin (Fig. 4a). The one luminal lysine residue of Synaptogyrin-1 (K136) was found to be labelled (see above) and cross-linked with luminal lysine residues of Synaptophysin (Fig. 4b). Studying Synaptophysin in detail, we found that many amino acids of both luminal loops were modified by the two labelling reagents. The lysine residues located in these loops (K83, K163, K173 and K186) heavily cross-link together (Fig. 4c). Three cross-links (Synaptophysin$^{K83}$-Synaptophysin$^{K83}$, Synaptophysin$^{K186}$-Synaptophysin$^{K186}$ and Synaptophysin$^{Nterm}$-Synaptophysin$^{Nterm}$) contain overlapping peptide sequences and therefore originate from two copies of the protein indicating formation of Synaptophysin dimers as also observed from western blotting (Fig. 4, Supplementary Fig. 6 and Supplementary Data 3). Note that previously described co-elution and non-covalent gas-phase association of peptides[52] can be excluded for these peptides as linear and cross-linked peptide pairs have been separated during the enrichment step (Methods). Cross-links identified within the same protein but originating from two copies of an oligomer cannot be distinguished by their mass spectra. However, intra-molecular cross-links identified in the luminal loops of Synaptophysin (Synaptophysin$^{K83}$- Synaptophysin$^{K163/173/186}$ and Synaptophysin$^{K186}$-Synaptophysin$^{K163/173}$) also support dimerisation through inter-molecular interactions (Fig. 4c). Note that no cross-links were identified for the large and flexible N-terminus of Synaptophysin because a very limited number of lysine residues is available. In addition, interactions between Synaptophysin and Synaptoporin (an isoform of Synaptophysin) are also mediated through the luminal loops (Fig. 4c). Taken together, our results show that the large and flexible loops of SV2A, Synaptophysin, Synaptoporin and Synaptogyrin-1 are involved in complex

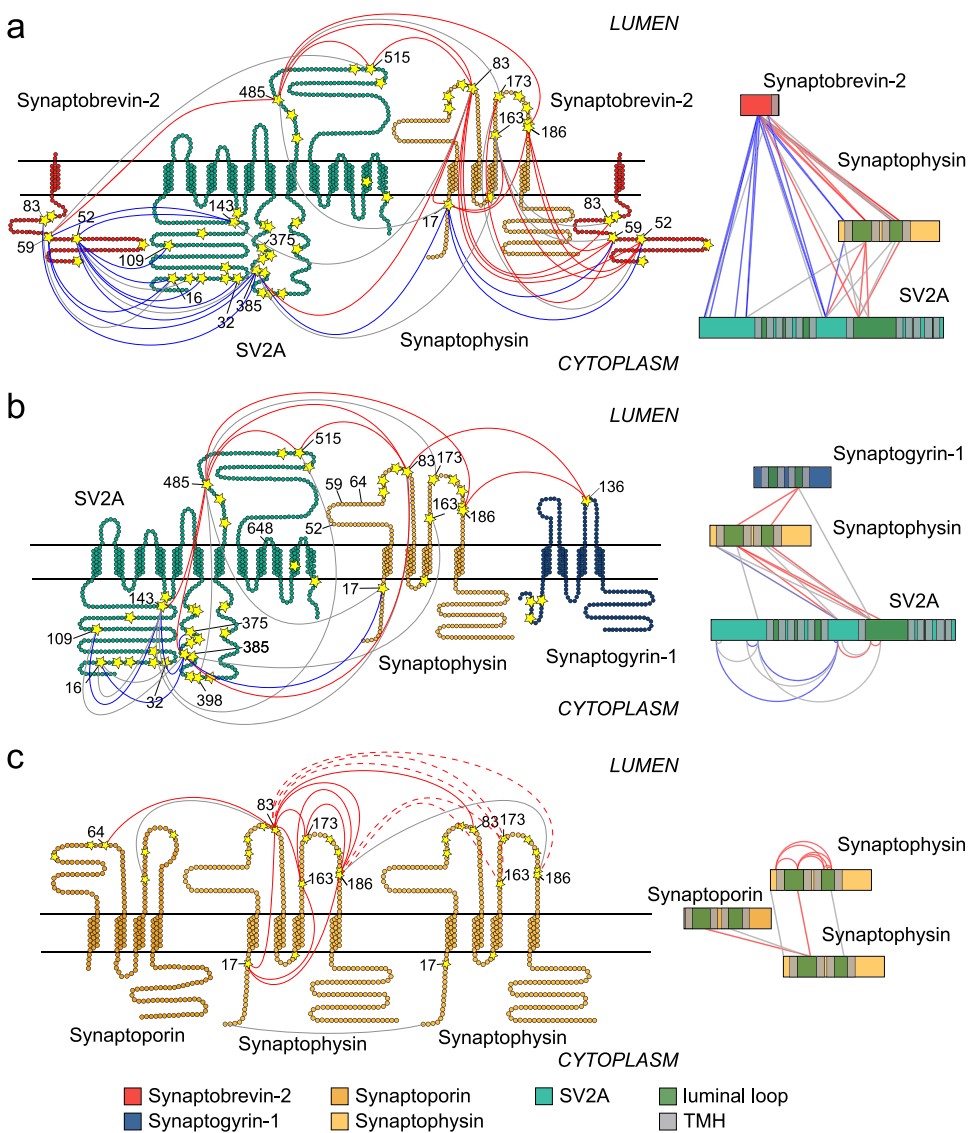

**Fig. 4 Protein interactions identified in flexible, luminal loops.** Topology models are shown for Synaptobrevin-2 (red), Synaptoporin/Synaptophysin (yellow/orange), SV2A (green) and Synaptogyrin-1 (blue). Lysine residues are labelled with residues numbers. Cross-links are indicated by solid and dashed lines (across the membrane and luminal, red; cytosolic, blue; interactions identified in only one replicate, grey). Chemically labelled amino acid residues are indicated by stars. Interactions between individual proteins are shown in network plots (rhs). The length of the bars represents the protein length. The N-terminus is on the left and the C-terminus is on the right side of the bar. Transmembrane helices (TMH) and luminal loops are indicated on the protein bars. **a** Interactions and accessible amino acid residues of Synaptobrevin-2, SV2A and Synaptophysin. **b** Interactions and accessible amino acid residues of SV2A, Synaptophysin and Synaptogyrin-1. **c** Interactions and accessible amino acid residues of Synaptophysin and Synaptoporin. Cross-links that support oligomerisation of Synaptophysin are shown as dashed lines.

formation in synaptic vesicles. Their accessibility for labelling and cross-linking reagents together with identified cross-links between cytosolic and luminal amino acids suggest an arrangement of these loops that makes them accessible from the cytosol thereby allowing protein interactions as well as dimerisation of Synaptophysin.

**Targeting Synaptobrevin-2 in its native environment.** The interaction network obtained from chemical cross-linking suggests that Synaptobrevin-2 plays a central role in complex formation in purified, unstimulated synaptic vesicles used here. There are two possible scenarios: first, due to its high abundance in synaptic vesicles[15], Synaptobrevin-2 is specifically involved in many protein interactions, and second, Synaptobrevin-2 non-specifically contacts many vesicle proteins induced by its high

structural dynamics reported previously[53–55]. To answer this question, we followed three approaches: (i) We cleaved Synaptobrevin-2 with BoNT B (see above) to remove the majority of the cytosolic domain from the vesicle membrane, (ii) we engaged Synaptobrevin-2 in SNARE complex formation by addition of the soluble ΔN-SNARE complex and (iii) we fused synaptic vesicles with 'empty' liposomes to provide a larger membrane for the vesicle proteins. Subsequently, we chemically cross-linked the vesicle proteins as described above and investigated protein interactions.

We first studied protein interactions in the absence of Synaptobrevin-2. For this, we chose a concentration of BoNT B at which Synaptobrevin-2 was completely cleaved (compare Supplementary Fig. 7). After successful cleavage of Synaptobrevin-2, proteins of synaptic vesicles were cross-linked with BS3 followed by LC-MS/MS analysis of covalently linked tryptic

peptides as described above. In total, we identified 216 cross-links in three biological replicates corresponding to 186 intra-molecular and 30 inter-molecular cross-links (Supplementary Data 6). Of these, 159 cross-links were identified in at least two replicates (Supplementary Fig. 8). Even though the number of identified cross-links is reduced in this experiment, we obtained a high number of fragment mass spectra for all individual cross-links confirming an overall sufficient LC-MS/MS analysis. Most of the proteins that were found to be cross-linked after BoNT B cleavage are contaminants (Supplementary Data 6). Nonetheless, several synaptic vesicle proteins that have been identified in our initial interaction network are also involved in the interactions identified in this experiment. These include Synaptobrevin-2, Synaptogyrin-1, CSP, Synaptophysin, SV2A and SV2B (Supplementary Fig. 9). As expected, the interaction network shows that the number of interactions with Synaptobrevin-2 is heavily reduced; only two cross-links were identified between Synaptobrevin-2 and other proteins. Importantly, interactions within the V-ATPase complex are comparable with those identified in untreated synaptic vesicles (Supplementary Fig. 9) confirming the applicability of cross-linking to BoNT B treated vesicles. The AP2 complex, on the other hand, does not make inter-molecular contacts after BoNT B treatment and only intra-molecular cross-links are observed for AP2 subunits (Supplementary Data 6). We further observed interactions between individual synaptic vesicle proteins that have also been identified before, for instance between Synaptophysin and V-ATPase-a1, SV2A or Synaptogyrin-1 as well as the CSP multimer. While we realise that cleavage of Synaptobrevin-2 strongly affects the protein interactions in synaptic vesicles in general, these protein interactions appear to be stable protein assemblies in synaptic vesicles.

We next assembled Synaptobrevin-2 in the SNARE complex and thereby locked the protein in a post-fusion state. For this we incubated synaptic vesicles with the so-called soluble ΔN-SNARE complex. The soluble ΔN-SNARE complex contains full-length SNAP25 (residues 1–206), the SNARE domain of Synatxin-1A (residues 188–259) as well as a C-terminal peptide of the soluble domain of Synaptobrevin-2 (residues 49–96). The pre-assembled complex forms a four-helix bundle as observed for the SNARE complex during membrane fusion[22,56,57]. Due to the missing N-terminus of Synaptobrevin-2, the full-length protein anchored in the vesicular membrane can assemble into the ΔN-SNARE complex and, as a consequence, the C-terminal Synaptobrevin-2 peptide dissociates. As Synaptobrevin-2 cannot be cleaved by BoNT B when assembled in the SNARE complex, we made use of BoNT B to verify complete engagement of Synaptobrevin-2 in the SNARE complex (Supplementary Fig. 10A).

We then used chemical cross-linking to identify protein interactions in synaptic vesicles incubated with the ΔN-SNARE complex. Due to the high abundance of the three SNARE proteins, we extended the LC gradient to allow more analysis time for selection of low abundant cross-linked species. Using this optimised method, we identified and verified 407 cross-links corresponding to 260 intra-molecular and 147 inter-molecular cross-links (Supplementary Data 7). 226 of these cross-links were identified in at least 2 biological replicates (Supplementary Fig. 11). The resulting interaction network (Supplementary Fig. 10b) reveals many interactions between the SNAREs SNAP25, Syntaxin-1A and Synaptobrevin-2 confirming formation of the SNARE complex. Note that cross-linked residues of Synaptobrevin-2 correspond to the soluble, C-terminal peptide and, therefore, these interactions most likely originate from an excess of ΔN-SNARE complex which was added to synaptic vesicles. Again, interactions within the V-ATPase complex are comparable to those identified above. However, when comparing interactions of individual proteins observed in this experiment

with those identified in untreated synaptic vesicles, the number of cross-links between Synaptobrevin-2 and other proteins is reduced. Instead, we observed many interactions between the SNARE proteins SNAP25 or Syntaxin-1A and other proteins including both synaptic vesicle components as well as contaminants (Supplementary Fig. 10B). We assume that the SNARE complex which assembled on vesicular Synaptobrevin-2 now captures the previously identified cross-links in the crowded environment of synaptic vesicles. Most frequent are interactions between the SNAREs and V-ATPase-a1 or Synapsin-1. Several binary interactions, for instance between Synaptobrevin-2, Synaptophysin and SV2A as well as cross-links within the Synaptophysin and Synaptobrevin-2 dimers/multimers are retained in these experiments suggesting that these proteins form stable interaction modules. Importantly, for Synaptotagmin-1, which was underrepresented in complex formation in unstimulated vesicles (Fig. 3), several cross-links with the SNARE complex were identified. Interactions between Synaptotagmin-1 and the SNARE complex were previously described in primed pre-fusion complexes[58] and might also form in this activated state. In addition, we observed multiple cross-links between the SNAREs and Rab1A, Rab3A and Rab39A proteins for which only intra-molecular cross-links were observed in unstimulated synaptic vesicles (Supplementary Data 3). As Rab proteins are involved in trafficking and vesicle recycling[59,60], we assume that these interactions resemble pre- and post-fusion states.

**Fusion of synaptic vesicles and liposomes reveals stable protein interactions**. Having targeted Synaptobrevin-2 directly, we next fused synaptic vesicles with empty liposomes thereby providing the vesicle proteins with more membrane space. For this, the ΔN-SNARE complex, i.e. a complex containing full-length SNAP25, the membrane-anchored Syntaxin-1A SNARE motif (residues 183–288) and the C-terminal peptide of the soluble domain of Synaptobrevin-2 (residues 49–96), was incorporated into liposomes and incubated with purified synaptic vesicles. Full-length Synaptobrevin-2 present in the synaptic vesicle membrane assembles with the ΔN-SNARE complex and thereby induces membrane fusion. To verify successful membrane fusion, we followed the fusion process by monitoring the size of vesicular species at different time-points using dynamic light scattering. For better distinction between synaptic vesicles, liposomes and fused vesicles, we used large ΔN-SNARE-proteoliposomes of 400 nm. Indeed, we observed a shift of the size distribution to large fused vesicles after 60–90 min (Supplementary Fig. 12). To verify that only few copies of Synaptobrevin-2 assemble into the SNARE fusion machinery and the majority of copies is available for complex formation, we again made use of BoNT B. Complete cleavage of Synaptobrevin-2 confirmed its availability in fused synaptic vesicles (Supplementary Fig. 13). Synaptic vesicle proteins were then cross-linked to study their interactions in this less crowded environment. We identified and verified 571 cross-links (440 intra- and 131 inter-molecular; Supplementary Data 8) of which 278 cross-links were identified in at least two replicates (Supplementary Fig. 14). Again, cross-links observed for the V-ATPase complex are comparable to those observed in untreated vesicles. In contrast to untreated synaptic vesicles, we do not observe the Synaptobrevin-2-centered interaction network (Fig. 5). Instead, interactions with specific proteins unveil. These are involved in smaller networks between Synaptobrevin-2, Synaptophysin, V-ATPase-a1, SV2A and Synaptoporin. In addition, cross-links within the Synaptophysin and CPS dimers/multimers or between the three SNARE proteins were observed. Cross-links of low frequency were identified between Synaptobrevin-2 and Synaptotagmin-1 or Rab3A as well as between Synaptophysin and

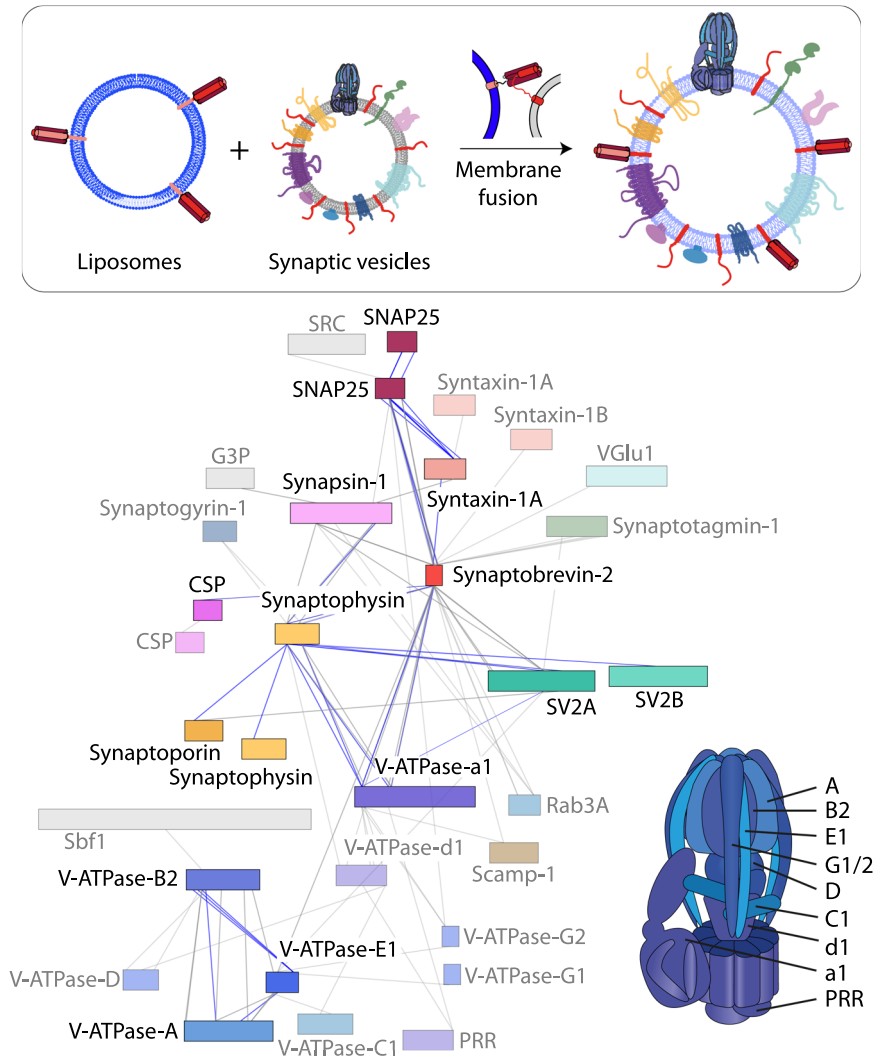

**Fig. 5 Protein interaction network of synaptic vesicles after fusion with ΔN-SNARE-proteoliposomes.** Top schematic: Synaptic vesicles were fused with ΔN-SNARE-proteoliposomes. During membrane fusion, vesicular Synaptobrevin-2 integrates into the SNARE complex. Fused vesicles provide a more spacious membrane environment. Network: Synaptic vesicle proteins are shown as coloured bars; contaminants are shown as grey bars. The length of the bars represents the protein length. The N-terminus is on the left and the C-terminus is on the right side of the bar. Inter-molecular cross-links identified in one (grey) or at least two (blue) biological replicates are shown. Proteins that are linked through interactions from only one biological replicate are transparent. A structural cartoon of the V-ATPase is shown for comparison. V-ATPase subunits identified in this interaction network are indicated.

Synaptogyrin-1. The absence of the large Synaptobrevin-2 network indicates that most of these interactions were rather unspecific and mainly form due to the high flexibility and abundance of Synaptobrevin-2. By allowing the proteins to spatially separate, we identify stable protein interactions which are maintained in the more spacious environment of fused synaptic vesicles and most likely contribute to the proteins' function.

## Discussion

We used chemical cross-linking and MS to uncover protein interactions in synaptic vesicles. Interactions between Synaptobrevin-2 and other proteins were remarkably abundant in purified, unstimulated vesicles suggesting a central role for Synaptobrevin-2 in complex formation. Inhibited and active states of the V-ATPase as well as association of the endocytotic AP2 complex were also observed. By cross-linking differently treated vesicles, we were able to distinguish unspecific interactions that were formed due to 'crowding' conditions in the vesicle membrane from stable, specific interactions that are retained in a

more spacious membrane environment. These experiments therefore reveal interaction modules which likely form functional protein assemblies in synaptic vesicles including interactions of Synaptobrevin-2, Synaptophysin, SV2A (SV2B) and V-ATPase-a1 as well as multimers (dimers) of Synaptobrevin-2, CSP and the two isoforms Synaptophysin and Synaptoporin.

From a technical point of view, chemical cross-linking uncovered, protein interactions in synaptic vesicles in an unbiased manner, i.e. we were able to identify protein interactions without the need for specific antibodies or detergent-based protein solubilisation. The latter is of particular importance, as detergent solubilisation yielded complexes differing in composition[29,30] suggesting that the protein complexes of synaptic vesicles are sensitive towards the detergent used. In addition to simple protein-protein interactions, our approach delivers detailed insights into the interactions formed at residue-residue resolution. Note, that the latter depends on the cross-linking chemistry applied, for instance the presence of lysine residues when using the BS3 cross-linker. In addition, one should keep in mind that cross-linking only reveals binary interactions and an interaction network might contain

various protein complexes. The assembly of ternary complexes or larger assemblies, therefore, has to be confirmed through other approaches, for instance western blotting of cross-linked complexes or pull-down experiments. This is particularly true for interaction networks identified in synaptic vesicles as they pass through different functional states of the vesicle cycle and the assemblies fulfil the required functions at the various stages. Nonetheless, together with additional biochemical approaches and the findings of previous studies, we were able to distinguish different sub-populations of synaptic vesicles corresponding to the different states of the synaptic vesicle cycle and to define stable interaction modules. As mentioned above, unspecific cross-linking might be induced by proximity of proteins in a crowded environment such as that of synaptic vesicle membranes. These interactions cannot be distinguished from specific interactions of tightly bound proteins. To overcome crowding in the vesicle membrane, we followed different approaches and, importantly, obtained a protein interaction network in a more spacious membrane environment where cross-linking due to crowding in the membrane can be neglected. In summary, we made the following observations:

Protein interactions corresponding to different states or sub-populations of synaptic vesicles were identified. In particular, cross-links identified in the V-ATPase complex provided insights into the regulation of neurotransmitter loading. Previous studies proposed that one or two V-ATPase complexes are anchored to the vesicle membrane[15,20]. Our cross-linking experiments, however, revealed interactions that exclusively satisfy the fully assembled $V_1V_O$-ATPase or the dissociated $V_O$-ATPase membrane domain. Together with our proteomic analysis, which showed that $V_O$-ATPase subunits are more abundant than $V_1$-ATPase subunits, we suppose that either several V-ATPase complexes incorporate into the vesicle membrane during endocytosis and only a subset of vesicles maintains the $V_1$-domain after neurotransmitter loading or that a population of synaptic vesicles without $V_1$-ATPase complexes exists. These findings are supported by two recent studies showing that, first, $V_1/V_O$ assembly depends on the synaptic vesicle cycle and thereby modulates exocytosis[61] and, second, clathrin coated, endocytosed vesicles maintain their ability to acidify[62]. Most likely, synaptic vesicles purified from rat brain synaptosomes contain subsets of vesicle populations from different states of the synaptic vesicle cycle.

Apart from the V-ATPase complex, many proteins contribute to the interaction network of purified and unstimulated synaptic vesicles. While interactions with contaminants were identified in only one of four biological replicates, reproducible interactions dominate the network. However, due to the crowded environment in synaptic vesicle membranes, these interactions might be considered unspecific and additional experiments are required to distinguish specific from unspecific interactions. We therefore followed different biochemical and biophysical approaches. Specifically, we targeted interactions between Synaptobrevin-2 and ten synaptic vesicles protein markers as well as the subunits of the AP2 complex. We could show that these interactions are likely attributed to the high copy number of Synaptobrevin-2[15,21] as well as its inherent structural flexibility described previously[53–55]. Association of the soluble SNARE complex with Synaptobrevin-2 therefore induced many interactions between the SNARE complex and previous interaction partners of Synaptobrevin-2. Following BoNT B cleavage or fusion of the vesicles with liposomes, only a few of these interactions were maintained. In fact, cleavage of Synaptobrevin-2 with BoNT B caused significant disruption of inter-molecular protein interactions suggesting that Synaptobrevin-2 represents a hub in the protein assemblies of synaptic vesicles. The majority of these interactions, however, might be transient as indicated by their disappearance in a larger vesicle membrane.

The interaction network also revealed dimerisation (multimerisation) of Synaptobrevin-2, Synaptophysin and CSP. Multimers of different stoichiometries were described for Synaptobrevin-2 and Synaptophysin in early studies[23,26,63]. As cross-linking MS only reveals binary interactions (see above), our MS-based approach cannot distinguish between dimers or higher multimers. However, in agreement with our previous study combining cross-linking MS and native MS[53], we observed multimers of Synaptobrevin-2 of at least pentamers by western blotting of cross-linked synaptic vesicles (Supplementary Fig. 6). For Synaptophysin, a dimeric complex but no higher multimers were observed. Importantly, one cross-link between Synaptophysin and Synaptoporin (Synaptophysin$^{K83}$-Synaptoporin$^{K64}$) was identified in four biological replicates indicating that, in contrast to previous findings[63], these two isoforms specifically and reproducibly interact. This specific cross-link was formed between two lysine residues located in protein regions of sequence similarity. The same region was found to mediate dimer (multimer) formation of Synaptophysin (Fig. 4 and Supplementary Data 3). We therefore assume that Synaptoporin and Synaptophysin can form mixed dimers or multimers. However, even though sequence alignments of the two proteins reveal similarity in the respective regions[64], interactions between Synaptoporin and interaction partners of Synaptophysin were not observed suggesting that formation of these complexes is rather specific. A similar scenario was discovered for the related protein Synaptogyrin-1; while reproducible cross-links between Synaptophysin and Synaptogyrin-1 were identified, interactions between Synaptogyrin-1 and other proteins are rather infrequent. However, this might also be attributed to the low number of cross-linking and tryptic cleavage sites in the Synaptogyrin-1 sequence.

As also reported in previous studies, Synaptobrevin-2 and Synaptophysin form a stable complex in synaptic vesicles which was also maintained in a spacious membrane environment after fusion with liposomes (Figs. 3, 5). The functional role of this specific interaction was discussed before suggesting, for instance, a control mechanism for Synaptobrevin-2 to be released from the Synaptophysin complex and entering the SNARE complex when required[26] or contribution to the fusion pore by forming a hexameric complex of 2:1 Synaptobrevin-2:Synaptophysin subcomplexes[27]. We found that an excess of the soluble ΔN-SNARE complex as well as addition of BoNT B induce release of Synaptobrevin-2 from the complex supporting the proposed control mechanism of a pre-assembled complex mediated by dynamic interactions that allow disassembly and possibly reassembly of the two proteins. A weak signal observed by western blotting together with numerous interactions of Synaptobrevin-2 with other proteins indicate a relatively low abundance of this complex; however, its sensitivity against freezing has been described previously[29] and dissociation due to sample handling cannot be ruled out. The suggested 2:1 stoichiometry of the complex could not be confirmed in our experiments.

The inter-molecular interactions observed by chemical cross-linking reveal four ternary sub-networks (Fig. 3) which might or might not be part of the same protein complex. Importantly, these smaller networks are in whole or at least in part maintained after fusion of synaptic vesicles with 'empty' liposomes, i.e. these proteins associate even in a more spacious membrane environment and, therefore, can be considered specific interaction partners. Other protein interactions, observed in the crowded membrane of untreated synaptic vesicles, were not identified after membrane expansion and are rather unspecific. Two early studies used detergent solubilisation and immunoprecipitation or co-sedimentation to identify protein complexes of synaptic vesicles.

Interestingly, one of the previously observed complexes consists of Synaptobrevin-2, Synaptophysin and the $V_O$-ATPase[29] which together resemble one of the dominant sub-networks observed here. Other complexes were largely detergent-dependent and comprised Synaptobrevin-2, Synaptophysin, SV2 and in some cases Synaptotagmin-1 and/or the V-ATPase[30]. Again, these protein components were also linked predominantly in our interaction network while other major components of synaptic vesicles (e.g. Synaptotagmin-1) were underrepresented. We therefore conclude that the majority of interactions in synaptic vesicle membranes are formed by Synaptobrevin-2, Synaptophysin/Synaptoporin, SV2A (SV2B) and the V-ATPase complex.

Of these interactors, Synaptophysin/Synaptoporin, Synaptogyrin and SV2A/B are integral membrane proteins with four (Synaptophysin, Synaptoporin, Synaptogyrin) or twelve (SV2A/B) transmembrane helices connected through several large and flexible, luminal and cytosolic loops. Surprisingly, close inspection of the interactions formed between these proteins revealed extensive cross-linking through both luminal and cytosolic loops. These observations were confirmed by an independent labelling experiment employing non-membrane permeable chemical reagents. In addition, the integrity of purified synaptic vesicles and the correct protein orientation were confirmed by additional experiments including electron microscopy and a BoNT B cleavage assay. Furthermore, the vesicles used in our experiments proved functionally active as verified by a fusion assay (Supplementary Fig. 12). Therefore, the question how these interactions are accomplished remains. One possibility is the integration of luminal loops in the vesicular membrane making them accessible for non-membrane permeable cross-linking and labelling reagents. This assumption is underpinned by two previous findings: First, in phospholipid bilayers, Synaptophysin forms doughnut-shaped or rosette-like pores with a central, hollow cavity which accumulates uranyl salts in negative stain electron microscopy[25]. Structural similarity with cation channels suggests that large and dynamic loops might line the inner cavity of Synaptophysin multimers and are therefore exposed to the cytosol. Second, similar to viral capsids, which expand or shrink due to their requirements, synaptic vesicles were found to reversibly increase in size upon neurotransmitter loading. This increase is likely attributed to conformational rearrangements of SV2A and possibly other structurally similar synaptic vesicle proteins[65]. Our cross-links suggest that at least a population of synaptic vesicles undergoes a non-conventional structural rearrangement exposing luminal domains and making them accessible from the cytosol. This assumption is strongly supported by interactions between luminal loops of Synaptophysin and SV2A with cytosolic loops and Synaptobrevin-2. Cross-linking experiments further suggest that mixed multimers are formed from these structurally similar proteins. The functional role of Synaptophysin and related proteins has not yet been uncovered and suggestions range from fusion pore formation to regulation of exo- and endocytosis. Considering our findings, a role for Synaptophysin and SV2A, and possibly Synaptoporin and Synaptogyrin, in modulating synaptic vesicle size is hypothesised.

## Methods

**Purification of synaptic vesicles**. Synaptic vesicles were purified from rat brain as previously described[32,33]. The rats were kept according to the ethical guidelines approved by the Office of Veterinary Affairs and Consumer Protection of the city of Göttingen, Germany (permit number 32.22/Vo). Briefly, brain tissue was homogenised in 5 mM HEPES-KOH, pH 7.4 containing 320 mM sucrose, 0.2 mM phenylmethylsulfonyl fluoride, 1 µg/ml pepstatin, 0.1% (v/v) ethanol, 0.1% (v/v) DMSO. Cell debris was removed by centrifugation and synaptosomes were pelleted. Intact synaptosomes were washed and osmotically lysed with ice-cold water. Synaptic vesicles were pelleted by centrifugation and suspended in 5 mM

HEPES-KOH, pH 7.4 containing 40 mM Sucrose. The suspension was layered on top of a continuous 0.05-0.8 M sucrose gradient. After centrifugation, synaptic vesicles were collected at approximately 0.2–0.4 M sucrose and subjected to size-exclusion chromatography using controlled pore size glass beads for further purification. Synaptic vesicles were then concentrated by ultracentrifugation. The protein concentration was determined by UV spectroscopy at 280 nm using a DeNovix Spectrophotometer DS 11+. The typical protein concentration of purified synaptic vesicles was 2 µg/µl.

**Gel electrophoresis and western blotting**. Gel electrophoresis was performed using the NuPAGE system (Thermo Fisher Scientific) according to the manufacturer's protocols. Briefly, proteins were separated for 35 min at 200 V on 4–12% Bis–Tris or for 90 min at 125 V on 16% Tricine protein gels. Protein bands were stained with InstantBlue Coomassie Protein Stain solution (Expedeon). For western blotting, the proteins were separated by gel electrophoresis as described and transferred onto a nitrocellulose membrane (Roth) for 3 h at 50 mA. The membrane was blocked for 1 h with PBS/0.02% (v/v) Tween-20 containing 5% (m/v) milk powder and subsequently washed three times with PBS/0.02% (v/v) Tween-20. The membrane was incubated with anti-Synaptobrevin-2 clone 69.1 (1:10,000 in PBS/0.02% (v/v) Tween-20; SynapticSystems), anti-VAMP1/2/3 (1:50,000 in PBS/0.02% (v/v) Tween-20; SynapticSystems) or anti-Synaptophysin-1/2 (1:5,000 in PBS/0.02% (v/v) Tween-20; SynapticSystems) in PBS/0.02% (v/v) Tween-20/1% (m/v) BSA overnight. After three washing steps, the membrane was incubated with anti-mouse (for anti-Synaptobrevin-2 clone 69.1 and anti-Synaptophysin-1/2; 1:100,000 in PBS/0.02% (v/v) Tween-20/1% (w/v) BSA) or anti-rabbit (for anti-VAMP1/2/3; 1:100,000 in PBS/0.02% (v/v) Tween-20/1% (w/v) BSA) secondary antibody. Proteins were detected using the Pierce ECL Western Blotting Substrate (Thermo Fisher Scientific). Chemiluminescence was detected using the luminescent image analyzer LAS-4000 (Fujifilm Corporation).

**Sample preparation for protein identification**. The proteins of synaptic vesicles were separated by gel electrophoresis as described above. Entire gel lanes were cut into 23 equally sized gel slices using a gel cutter[66]. The proteins were then hydrolysed with trypsin as described previously[67]. For this, the proteins were first reduced with dithiothreitol and reduced cysteine residues were then alkylated with iodoacetamide. The proteins were hydrolysed with trypsin at 37 °C overnight. Extracted peptides were dried in a vacuum centrifuge and dissolved in 2% (v/v) acetonitrile/ 0.1% (v/v) formic acid for LC-MS/MS analysis.

**Chemical cross-linking**. For chemical cross-linking of synaptic vesicle proteins, synaptic vesicles (protein concentration 2 µg/µg) were incubated with 10 mM BS3 for 1 h at 25 °C and 300 rpm in a thermomixer. The proteins were then precipitated with 1 vol. ice-cold ethanol and 1/10 vol. 3 M sodium acetate, pH 5.3 at −20 °C. The protein pellet was washed with 80% (v/v) ice-cold ethanol and dried in a vacuum centrifuge.

For tryptic hydrolysis, the protein pellet was dissolved in 20 µl 1% (m/v) RapiGest (Waters) in 25 mM ammonium bicarbonate, pH 8.5. Proteins were then reduced with 20 µl 50 mM dithiothreitol in 25 mM ammonium bicarbonate, pH 8.5 for 1 h at 37 °C. Reduced cysteines were alkylated with 20 µl 100 mM iodoacetamide in 25 mM ammonium bicarbonate, pH 8.5 for 1 h at 37 °C in the dark. For tryptic digestion, the concentration of RapiGest was diluted to 0.1% (m/v) with 25 mM ammonium bicarbonate, pH 8.5. Trypsin (Promega) was added at a 1:20 enzyme: protein ratio followed by incubation at 37 °C overnight. RapiGest was then degraded by addition of 40 µl of 5% (v/v) trifluoric acid followed by incubation for 2 h at 37 °C. Degraded RapiGest was removed by centrifugation and tryptic peptides were dried in a vacuum centrifuge.

Cross-linked peptide pairs were enriched by peptide size-exclusion chromatography on an Superdex™ peptide 3.2/300 column using an ÄKTA pure chromatography system (GE Healthcare). For this, the peptides were dissolved in 30% (v/v) acetonitrile, 0.1% (v/v) trifluoroacetic acid followed by isocratic separation at a flow rate of 50 µl/min. Elution of the peptides was monitored at 280 nm. Fractions of 50 µl were collected. Fractions containing cross-linked peptide pairs as well as linear peptides were dried in a vacuum centrifuge and dissolved in 2% (v/v) acetonitrile/ 0.1% (v/v) formic acid for LC-MS/MS analysis.

**Chemical labelling with sulfo-NHS-acetate and DEPC**. For labelling with sulfo-NHS-acetate, a total protein amount of approximately 30 µg was incubated with 5 and 10 mM sulfo-NHS-acetate for 15 min at 23 °C. The proteins were then precipitated with 1 vol. ice-cold ethanol and 1/10 vol. 3 M sodium acetate, pH 5.3 at −20 °C. The protein pellet was washed with 80% (v/v) ice-cold ethanol and dried in a vacuum centrifuge. The proteins were digested with RapiGest as described for cross-linked proteins (see above). The peptides were dissolved in 2% (v/v) acetonitrile/0.1% (v/v) formic acid for LC-MS/MS analysis.

For labelling with DEPC, a total protein amount of approximately 30 µg was incubated with 5 and 10 mM DEPC for 10 min at 37 °C. The proteins were then precipitated with 1 vol. ice-cold ethanol and 1/10 vol. 3 M sodium acetate, pH 5.3 at −20 °C. The protein pellet was washed with 80% (v/v) ice-cold ethanol and dried in a vacuum centrifuge. The protein pellet was dissolved in 8 M urea in 25 mM Tris-HCl, pH 7.9 followed by incubation for 15 min at room temperature. The

proteins were reduced with 5 mM dithiothreitol for 30 min at 23 °C followed by alkylation with 20 mM iodoacetamide for 30 min at 23 °C. The solution was then diluted with 100 mM ammonium bicarbonate, pH 8.5 to a final concentration of 2 M urea. Tryptic hydrolysis was performed at 37 °C overnight (enzyme:protein 1:20). Generated peptides were desalted using ZipTips (Merck Millipore). Briefly, the C18 material of the ZipTips was equilibrated with 60% (v/v) acetonitrile/0.05% (v/v) formic acid and adjusted to loading conditions with 0.05% (v/v) formic acid. The peptides were loaded onto the ZipTips and washed with 0.025% (v/v) FA. The peptides were eluted with 60% (v/v) ACN/0.1% (v/v) FA. The peptides were dried in a vacuum centrifuge and dissolved in in 2% (v/v) acetonitrile/ 0.1% (v/v) formic acid for LC-MS/MS analysis.

**LC-MS/MS**. Tryptic peptides were separated by nano-flow reversed-phase liquid chromatography (DionexUltiMate 3000 RSLCnano System, Thermo Scientific; mobile phase A, 0.1% (v/v) formic acid (FA); mobile phase B, 80% (v/v) acetonitrile (ACN)/ 0.1% (v/v) FA) coupled with a Q Exactive Plus Hybrid Quadrupole-Orbitrap mass spectrometer (Thermo Scientific) or an Orbitrap Fusion Tribrid mass spectrometer (Thermo Scientific). The peptides were loaded onto a trap column (μ-Precolumn C18 PepMap 100, C18, 300 μm I.D., particle size 5 μm; Thermo Scientific) and separated with a flow rate of 300 nL/min on an analytical C18 capillary column (50 cm, HPLC column Acclaim® PepMap 100, C18, 75 μm I.D., particle size 3 μm; Thermo Scientific). For protein identification (in-gel hydrolysis), a gradient of 4–90% (v/v) mobile phase B over 62 min was used. For chemically modified peptides (in-solution hydrolysis), a gradient of 4–90% (v/v) mobile phase B over 271 min was used. Cross-linked peptides were separated by a gradient of 4–90% (v/v) mobile phase B over 92 min (Q Exactive Plus) or 79 min (Orbitrap Fusion). For analysis of cross-linked peptides of synaptic vesicles incubated with the soluble ΔN SNARE complex, the gradient was extended to 152 min. The gradient was adjusted step-wise for early, middle and late fractions from peptide size-exclusion chromatography. The peptides were directly eluted into the mass spectrometer.

Typical mass spectrometric conditions for the Q Exactive Plus mass spectrometer were: spray voltage, 2.8 kV; capillary temperature, 275 °C; data-dependent mode. Survey full scans were acquired in the Orbitrap (m/z 350–1600) at a resolution of 70,000 and an automatic gain control (AGC) target of 3e6. The 20 most intense ions with charge states of 2+ to 7+ were selected for HCD MS/MS fragmentation in the HCD cell at a resolution of 17,500 and an AGC target of 1e5. Normalised collision energy was set to 30%. The fixed first mass was 105.0 m/z. Detection in the HCD cell of previously selected ions was dynamically excluded for 30 s. For cross-linking analysis, doubly charged ions were also excluded. Internal calibration of the Orbitrap was performed using the lock mass option (lock mass m/z 445.12002[68]).

Typical mass spectrometric conditions for the Orbitrap Fusion mass spectrometer were: spray voltage, 2.0 kV; capillary temperature, 275 °C; data-dependent mode. Survey full scans were acquired in the Orbitrap (m/z 350–1600) at a resolution of 120,000 and an AGC target of 5e5. The 20 most intense ions with charge states of 3+ to 8+ were selected for HCD MS/MS fragmentation in the HCD cell at a resolution of 30,000 and an AGC target of 5e4. Normalised collision energy was set to 30%. The fixed first mass was 110.0 m/z. Detection in the HCD cell of previously selected ions was dynamically excluded for 30 s. Internal calibration of the Orbitrap was performed using the lock mass option (lock mass m/z 445.12002[68]).

**Data analysis**. For protein identification, raw data were searched against the uniprot database (taxonomy *rattus norvegicus*; 31,568 proteins; UniProt proteome ID: UP000002494; 26 Sep 2018) using MaxQuant v1.6.3.4[69] with the following database search settings: enzyme, trypsin; mass accuracy of precursor ions in main search, 4.5 ppm; mass accuracy of fragment ions in main search, 0.5 Da; number of allowed missed cleavages, 2; variable modifications, carbamidomethylation of cysteine and oxidation of methionine; quantification, iBAQ[34]; FDR, 1%. The database search results were further processed with Perseus v1.6.2.3. For this, typical contaminants as well as protein hits with MaxQuant scores <100 and those which were only identified by their modified peptides were removed from the results list. The median iBAQ intensities obtained for all proteins from all replicates were visualised in scatter plots. The 400 most abundant proteins were selected for a synaptic vesicle protein database.

For identification of modified sites from labelling experiments, raw data were searched against the generated database using MaxQuant v1.6.3.3[69]. Database search parameters were: enzyme, trypsin; mass accuracy for precursor ions in main search, 4.5 ppm; mass accuracy for fragment ions, 0.5 Da; maximum number of missed cleavages, 4; variable modifications, N-terminal acetylation, carbamidomethylation of cysteine, oxidation of methionine, acetylation of lysine, serine, threonine and tyrosine, carboethoxylation of histidine, lysine, cysteine, serine, threonine and tyrosine including neutral losses from histidine, lysine, serine and threonine, formyl-carboethoxylation of histidine, di-carboethoxylation of histidine and urethane-carboethoxylation of histidine including neutral loss (see ref. [49] for details); FDR, 1%. The 'match between runs' option was enabled. Modified sites with probability scores <0.75 and peptide scores <80 were discarded.

For identification of cross-linked peptides, raw data were searched against the generated database using pLink v2.3.2 with the following search parameters: enzyme, trypsin; maximum number of missed cleavages, 2; minimum peptide

mass, 600; maximum peptide mass, 6000; minimum peptide length, 4; maximum peptide length, 60; mass tolerance for precursor ions, 20 ppm; mass tolerance for fragment ions, 20 ppm; variable modifications, carbamidomethylation of cysteine and oxidation of methionine; fragmentation, HCD; cross-linker, BS3 (i.e. reactive towards lysine residues and protein N-termini); false-discovery rate, 0.05. Results tables were formatted using the CroCo software[70]. Potential cross-links were manually checked for spectral quality. For validation, fragment ion series of at least 4 adjacent fragments should be observed for both peptides with reasonable intensity. Only cross-links which were observed in at least two replicates and validated in at least one fragment spectrum were retained. Protein network plots were generated using XVis[71]. Cross-links of the V-ATPase complex were mapped into the atomic model of a high-resolution structured using Xlink Analyzer[72] and UCSF Chimera[73].

**Botulinum neurotoxin B cleavage**. For cleavage with BoNT B, 4 vol. of synaptic vesicles (2 μg/μl protein concentration) were incubated for 1 h at 37 °C with 1 vol. of the supernatant of a BoNT B secreting clostridium cell culture. Cleavage was verified by western blotting using the anti-VAMP1/2/3 antibody.

**Purification of the soluble ΔN-SNARE complex and binding to synaptic vesicles**. Proteins of the soluble ΔN-SNARE complex were expressed with an N-terminal His-tag in *E. coli* and purified by immobilised metal affinity chromatography (IMAC) and anion/cation exchange chromatography as described[74,75]. Briefly, all proteins were purified through their His-tags by IMAC using Nickel columns. His-tags were removed by Thrombin cleavage and subsequent reversed IMAC. The proteins were collected in the flow-through and further purified by anion (Syntaxin-1A(188-259) and SNAP25(1-206)) or cation (Synaptobrevin-2(49-96)) exchange chromatography. Note that all cysteine residues of SNAP25 were mutated to serine residues. Syntaxin-1A(188-259), SNAP25 (1-206) und Synaptobrevin-2(49-96) were incubated at a molar ratio of 1:1:2 for 48 h at 4 °C. The assembled complex was purified by anion exchange chromatography. An excess of purified ΔN-SNARE complex (approximately 3 copies of the soluble ΔN-SNARE complex per copy of Synaptobrevin-2) was then added to synaptic vesicles followed by incubation for 16 h at 4 °C.

**Preparation of ΔN-SNARE-proteoliposomes**. Syntaxin-1A (183-288), the cysteine-free variant of SNAP25 (1-206) and Synaptobrevin-2 (49-96) with N-terminal His-tags were co-expressed in *E. Coli* and purified as described[76]. Briefly, the ΔN-SNARE complex was purified by IMAC using a Nickel column. The His-tags of the proteins were removed by Thrombin cleavage overnight and subsequent reversed IMAC. The complex was collected in the flow-through and further purified by anion exchange chromatography. The purified ΔN-SNARE complex was stored in 20 mM HEPES containing 200 mM sucrose, 300 mM NaCl and 2% (m/v) n-Octyl-β-D-glucopyranoside.

Unilamellar liposomes containing DOPC, DOPS and cholesterol at a molar ratio of 26:4:10 in 20 mM MOPS, pH 7.4 containing 150 mM potassium D-gluconate were prepared by reverse-phase evaporation[77] followed by extrusion with 11 strokes through a polycarbonate membrane with a pore size of 400 nm for fusion assays and 21 strokes through a polycarbonate membrane with a pore size of 100 nm for cross-linking experiments.

Liposomes were mixed with the ΔN-SNARE complex in n-Octyl-β-D-glucopyranoside at a molar ratio of 1:500. The detergent was removed by dialysis in two steps against liposome buffer (see above) using Slide-A-Lyzer cassettes with a molecular weight cut-off of 2 kDa (Thermo Fisher Scientific, Waltham, MA). The first dialysis step was performed overnight at 4 °C with 2 g/L of adsorbent beads (SM-2-Bio-Beads; Bio-Rad, Hercules, CA). The second dialysis step was performed without adsorbent beads for approximately 4 h at room temperature.

**Fusion assay followed by dynamic light scattering**. Fusion of isolated synaptic vesicles with ΔN-SNARE-proteoliposomes was followed by dynamic light scattering using a DynaPro® instrument (Wyatt Technology). First, the size distribution of freshly isolated synaptic vesicles and ΔN-SNARE-proteoliposomes were recorded separately. Membrane fusion was then initiated by mixing 25 μl of synaptic vesicles with 25 μL of 5-times diluted ΔN-SNARE-proteoliposomes in a black quartz cuvette placed in the DynaPro® instrument. The mixture was incubated at 37 °C and size distributions were recorded at 15, 60 and 90 min. Ten sequential measurements of 5 s were performed at each time-point using a scattering angle of 90°. The data were processed using the Dynamics V6 software (Wyatt Technology). The radii and the size distributions were calculated with the regularisation algorithm as specified in the software.

**Electron microscopy**. Purified synaptic vesicles were bound to a glow discharged carbon foil covered grid. After fixation with 1% glutaraldehyde followed by staining with 1% uranyl acetate, the samples were evaluated at room temperature using a Talos L120C transmission electron microscope (Thermo Fisher, Eindhoven, The Netherlands).

**Reporting summary**. Further information on research design is available in the Nature Research Reporting Summary linked to this article.

## Data availability

Data supporting the findings of this manuscript are available from the corresponding author upon reasonable request. A reporting summary for this Article is available as a Supplementary Information file. All MS raw files were deposited to the ProteomeXchange Consortium (www.proteomexchange.org) via the PRIDE[78] partner repository with the dataset identifier PXD020859. Databases are available under the same dataset identifier. Source data are provided with this paper.

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

## Acknowledgements

The authors thank Henning Urlaub for providing access to the Orbitrap Fusion mass spectrometer. This work was funded by the Federal Ministry for Education and Research (BMBF, ZIK programme, 03Z22HN22), the European Regional Development Funds (EFRE, ZS/2016/04/78115) and the MLU Halle-Wittenberg. MG was supported by a CAPES-Humboldt fellowship from the Alexander von Humboldt Foundation and from the Coordenaçao de Aperfeiçoamento de Pessoal de Nivel Superior (CAPES, process number 99999.002062/2014-03).

## Author contributions

S.W. performed cross-linking experiments and data analysis; M.B. performed mapping of V-ATPase cross-links; M.G. and S.K. purified synaptic vesicles; S.K. and S.W. performed western blots; S.W. and S.K. performed protein identification; M.G. and S.W. performed fusion experiments; M.B. performed labelling experiments and data analysis; D.R. performed electron microscopy of purified synaptic vesicles; A.P.L. purified the soluble ΔN-SNARE complex; R.J. and C.S. designed experiments; C.S. analysed the data, guided the research and wrote the manuscript with input from all authors.

## Funding

## Competing interests

The authors declare no competing interests.
