## [Peer Review File · Nature Communications]

Reviewer #1 (Remarks to the Author):

In this manuscript, Schmidt and co-workers apply cross-linking MS in combination with other proteomics methods to study protein interactions in synaptic vesicles. The authors first generate an inventory of the proteins contained in their synaptic vesicle preparations, which is used for generating databases for cross-linking experiments. This data is also used to confirm the subunit stoichiometry of the V-ATPase complex, which was found to be in agreement with previous data. In the following, synaptic vesicles were cross-linked using the non-permeable reagent BS3 and the identified cross-links were used to generate an interaction network. Cross-links on the V-ATPase were found to represent two functional states, and the XL-derived interaction network was found to feature Synaptobrevin-2 as a central hub protein. Because cross-links were identified between cytosolic and luminal loop regions on some proteins, sample integrity was evaluated using confirmatory experiments, proving indeed that such interactions are possible in intact vesicles. The central role of Synaptobrevin-2 was further explored by (1) functional assays, including in situ cleavage of its cytosolic domain and scavenging of the protein by recruitment to the SNARE complex, and (2) "diluting" the protein in expanded vesicles obtained by fusion with liposomes. Both scenarios resulted in a significant rearrangement of the XL interaction network.

The authors have paid a lot of attention to details and have thoroughly designed confirmation experiments, making the conclusions overall convincing despite the relatively low reproducibility of the cross-links in this challenging matrix. Examples for this include the observation of cross-links between cytosolic and luminal loops, which was followed up by Botulinium neurotoxin B cleavage assays and covalent labeling coupled to MS; and the study of concentration-dependent interaction networks involving Synaptobrevin-2 using fused synaptic vesicles.

The manuscript is written clearly and all results are presented in sufficient detail, including all mass spectrometry results. Raw data has additionally been deposited with the proteomeXchange/PRIDE repository.

Note that I cannot fully assess the relevance of the work for synaptic biology. From a methodological point of view, I have only some minor comments that should be addressed, which are summarized below:

Page 5, line 124: "identified >1000 proteins with high scores". This is ambiguous, it would be better to refer to a given FDR, for example.

Page 6, When discussing the elucidation/confirmation of V-ATPase subunit stoichiometry on page 6, the inhomogeneous distribution of trypsin cleavage sites caused some problems. Would a different protease have been more suitable for this experiment?

Page 8, line 184 and elsewhere: I suggest to avoid the term "di-peptide" and use "peptide pair" instead, because dipeptide may refer to a two amino acid-long peptide.

Page 25, line 652 and elsewhere: "trifluoric" probably means "trifluoroacetic"?

page 27-28: Some information about MS/MS settings is missing (collision energy, resolution). Why was the fragment accuracy set to 0.5 Da if high-resolution MS/MS detection was used?

reviewed by Alexander Leitner, ETH Zurich, Switzerland

Reviewer #2 (Remarks to the Author):

Wittig and colleagues use purified synaptic vesicles (SV) and a systematic cross-linking / MS approach to identify intra- and intermolecular protein-protein interactions on the vesicle surface. Importantly, they employ different conditions such as BoNTB cleavage of synaptobrevin-2, or incubation with the soluble domains of t-SNARE or reconstituted t-SNAREs to distinguish between stage specific interactions of synaptobrevin-2 and "random collision / crowding" binding. In addition, the results confirm the existence of two distinct states of the V1V0ATPase – fully assembled and

dissociated V0 membrane domain. Remarkably, despite the use of membrane impermeable crosslinkers, the authors observe cross-links between “luminal” protein domains and cross-links between cytosolic and luminal protein domains. This result is perplexing, but appropriate controls are presented suggesting that the synaptic vesicles were intact. The observation may suggest that the topology of selected multiple membrane spanning protein may be less defined or more flexible than expected. The authors propose that this flexibility may contribute to rosette-like pore formation (synaptophysin) or vesicle size dynamics. Overall, the paper is well written, the data is solid and adds some new insights to our understanding of SV protein networks. The following concerns should be addressed:

- Figure S1 indicates that the preparations also contain vesicle clusters. Can the authors exclude that some of the cross-links are generated by trans-interactions of proteins between apposing vesicles. Does dilution of the SV preparation change vesicle “aggregation” and the cross-link pattern?

Minor points:

- Protein patterns in Figure S2A seem to differ to some degree, e.g. preparations 3-5 contain a prominent band(s) around 70 kDa. Are these bands contaminations?
- Figure 4C: The authors mention that “Intra-molecular cross-links that also support oligomerisation of Synaptophysin are shown as dashed lines.” The dashed lines show inter-molecular cross-links. Please clarify or change wording.
- At the end of page 14 the authors state: “We next engaged Synaptobrevin-2 in SNARE complex formation and thereby locked the protein in a functionally active form.” To which degree are the SNARE motifs assembled in this complex? Does the complex indeed reflect an activated (prefusion) stage? The authors may consider to change the wording.

Reviewer #3 (Remarks to the Author):

The authors performed a proteomics and crosslinking mass spectrometry study on synaptic vesicles. While I can comment on the quality of the crosslinking mass spectrometry (XL-MS) findings and how to interpret them I cannot comment totally on the biological relevance and novelties of these results. The way the results are currently presented however, I fear their usefulness may be limited. Several replicated preps of the vesicles are performed and proteomics confirms them to resemble previous work confirming a good prep. Negative stain EM also confirms that these vesicles seem intact. The paper begins with a proteomics study of vesicles and while iBAQ is not absolute quantitation the estimation of relative abundances is enough for the presented study. I find this to be very thorough.

The XL-MS study was also competently done. It is stated that crosslinks were validated at 5% FDR using Plink but then a manual validation was performed. Why? How many crosslinks were removed manually?

My major concerns around information that can be gleaned from this study are as follows:

- The description of the crosslinked interaction network beings with V-type ATPase. I find the cartoon representation of the structure to be unsatisfactory. There has recently been a structure of a mammalian V-type ATPase solved and a homology model should be produced based on this and the crosslinks mapped on. This will vastly improve any structure-based arguments about the populations of V1-V0 and V0 in the sample. All of the structure-based arguments are weak until this is done. This should identify a cluster of unsatisfied links on the whole V0-V1 complex that can only be explained by a lone V0 structure, perhaps using DisVis if a suitable structure is not known.
- The authors convincingly demonstrate that the Synaptobrevin-2 crosslinks are difficult to interpret. Synaptobrevin-2 is a flexible protein that is found in relatively high concentration on these vesicles. The very long half-life of BS3 means that it will very often crosslink uncommon interactions of such a

protein. It was shown nicely in this work by addition of SNARE, dilution of vesicle surface or cleavage of the Synaptobrevin-2 that many of these detected interactions were such crosslinking-induced artifacts.

However, as many of the interactions identified here have already been shown in some form by other previous studies it would have been wise if the authors had used the structural information present in the crosslinking data to draw some structural conclusions. One example would be, if the interactions between the V0 region of the ATPase and Synaptobrevin-2 is specific there should be a binding 'site' that could be suggested by the crosslinks.

Other points.

Figure S4 shows spectra. My concern with this panel is that it is potentially simply two gas phase associated peptides instead of a real crosslinked di-peptide. The lack of b-ion fragment series concerns me. To check this, interrogate the spectra with one each of the peptides instead containing a monolink (hydrolysed crosslinker at the lysine) and see if this allows matching of b-ions to the two linear peptides.

Figure S5, why were vesicles crosslinked with 15mM BS3 for Western blot when the XL-MS was performed on 10mM crosslinked vesicles?

The meaning of the following sentence is unclear and should be clarified, "We therefore conclude that the main interactors of synaptic vesicles are Synaptobrevin-2, Synaptophysin/Synaptoporin, SV2A (SV2B) and the V-ATPase complex."

REVIEWER COMMENTS

Reviewer #1 (Remarks to the Author):

In this manuscript, Schmidt and co-workers apply cross-linking MS in combination with other proteomics methods to study protein interactions in synaptic vesicles. The authors first generate an inventory of the proteins contained in their synaptic vesicle preparations, which is used for generating databases for cross-linking experiments. This data is also used to confirm the subunit stoichiometry of the V-ATPase complex, which was found to be in agreement with previous data. In the following, synaptic vesicles were cross-linked using the non-permeable reagent BS3 and the identified cross-links were used to generate an interaction network. Cross-links on the V-ATPase were found to represent two functional states, and the XL-derived interaction network was found to feature Synaptobrevin-2 as a central hub protein. Because cross-links were identified between cytosolic and luminal loop regions on some proteins, sample integrity was evaluated using confirmatory experiments, proving

indeed that such interactions are possible in intact vesicles. The central role of Synaptobrevin-2 was further explored by (1) functional assays, including in situ cleavage of its cytosolic domain and scavenging of the protein by recruitment to the SNARE complex, and (2) "diluting" the protein in expanded vesicles obtained by fusion with liposomes. Both scenarios resulted in a significant rearrangement of the XL interaction network.

The authors have paid a lot of attention to details and have thoroughly designed confirmation experiments, making the conclusions overall convincing despite the relatively low reproducibility of the cross-links in this challenging matrix. Examples for this include the observation of cross-links between cytosolic and luminal loops, which was followed up by Botulinium neurotoxin B cleavage assays and covalent labeling coupled to MS; and the study of concentration-dependent interaction networks involving Synaptobrevin-2 using fused synaptic vesicles.

The manuscript is written clearly and all results are presented in sufficient detail, including all mass

spectrometry results. Raw data has additionally been deposited with the proteomeXchange/PRIDE repository.

Note that I cannot fully assess the relevance of the work for synaptic biology. From a methodological point of view, I have only some minor comments that should be addressed, which are summarized below:

Page 5, line 124: "identified >1000 proteins with high scores". This is ambiguous, it would be better to refer to a given FDR, for example.

An FDR of 1% is a default parameter of MaxQuant and was also applied in our analysis. We have added this information to the methods and results sections. In addition to a 1% FDR, we also applied a score cut-off to ensure high confidence and reduce the number of protein hits.

Page 6, When discussing the elucidation/confirmation of V-ATPase subunit stoichiometry on page 6, the inhomogeneous distribution of trypsin cleavage sites caused some problems. Would a different protease have been more suitable for this experiment?

Most of the typical synaptic vesicle components have large cytosolic domains or are soluble proteins that associate with the membrane. We used trypsin because this is well-suited to study most of these proteins. The V-ATPase is a specific case as it contains a large(er) membrane-embedded domain. Chymotrypsin would have been an alternative option, however, chymotryptic peptides are more difficult to ionise and the yield of peptides was significantly lower in our preliminary experiments. In our analyses we had to find the best option to identify as many proteins as possible and found that Trypsin outnumbered Chymotrypsin in terms of protein IDs and identified cross-links.

In addition, the identification of some V-ATPase subunits is difficult and most studies do not identify the full cohort of subunits. Our study provided for the first time a nearly complete set of subunits. (For comparison, see the recently published structure of the V-ATPase or previously published proteome studies as referenced in our manuscript.)

Page 8, line 184 and elsewhere: I suggest to avoid the term "di-peptide" and use "peptide pair" instead, because dipeptide may refer to a two amino acid-long peptide.

We agree and have changed the term to "cross-linked peptide pair" or "cross-linked peptide" as appropriate.

Page 25, line 652 and elsewhere: "trifluoric" probably means "trifluoroacetic"?
We have corrected this mistake.

page 27-28: Some information about MS/MS settings is missing (collision energy, resolution). Why was the fragment accuracy set to 0.5 Da if high-resolution MS/MS detection was used?

The collisional energy was given at the beginning of the section. We agree that this should be listed with the MS/MS parameters and have moved this information to the respective section. We have also included resolution of MS/MS spectra as well as first masses detected in MS/MS spectra.

reviewed by Alexander Leitner, ETH Zurich, Switzerland

Reviewer #2 (Remarks to the Author):

Wittig and colleagues use purified synaptic vesicles (SV) and a systematic cross-linking / MS approach to identify intra- and intermolecular protein-protein interactions on the vesicle surface.

Importantly, they employ different conditions such as BoNTB cleavage of synaptobrevin-2, or incubation with the soluble domains of t-SNARE or reconstituted t-SNAREs to distinguish between stage specific interactions of synaptobrevin-2 and “random collision / crowding” binding. In addition, the results confirm the existence of two distinct states of the V1V0ATPase – fully assembled and dissociated V0 membrane domain. Remarkably, despite the use of membrane impermeable crosslinkers, the authors observe cross-links between “luminal” protein domains and cross-links between cytosolic and luminal protein domains. This result is perplexing, but appropriate controls are presented suggesting that the synaptic vesicles were intact. The observation may suggest that the topology of selected

multiple membrane spanning protein may be less defined or more flexible than expected. The authors propose that this flexibility may contribute to rosette-like pore formation (synaptophysin) or vesicle size dynamics. Overall, the paper is well written, the data is solid and adds some new insights to our understanding of SV protein networks. The following concerns should be addressed:

- Figure S1 indicates that the preparations also contain vesicle clusters. Can the authors exclude that some of the cross-links are generated by trans-interactions of proteins between apposing vesicles. Does dilution of the SV preparation change vesicle “aggregation” and the cross-link pattern?

We agree with the reviewers that this is indeed a legitimate concern. However, there are different aspects that argue against this speculation. (i) The degree of vesicle clustering is low when compared with individual (non-clustered) vesicles. In general, cross-linked peptides are low-abundant; the abundance of cross-linked peptides originating from trans-vesicle interactions would be even lower so that an identification of these species would be rather impossible. (ii) The range of the cross-linking reagent is limited and the reactive groups of the cross-linker might not reach reactive amino acid side chains of other synaptic vesicles. (iii) Free reactive groups of the cross-linking reagent will be hydrolysed by the water molecules of the buffer resulting in a limited reaction time which further reduces this possibility.

Please also note that the synaptic vesicles have been diluted to the desired protein concentration before performing the cross-linking reaction.

Minor points:

- Protein patterns in Figure S2A seem to differ to some degree, e.g. preparations 3-5 contain a prominent band(s) around 70 kDa. Are these bands contaminations?

We agree that the protein patterns of the different preparations differ to some degree. This might have different reasons. First, the degree of contamination might be different. The individual preparations were performed from rat brain tissue on different days. The preparations therefore represent real biological replicates. This might cause differences in the contaminations. Second, note that there are some loading differences on the gels shown in Fig S2A. When studying intact organelles, it is difficult to determine the accurate protein concentration. The employed software (MaxQuant) is normalising these differences by clustering observed protein abundances. The prominent band at approx. 70 kDa might imply that the chaperone Hsp70 is present, however, it was not observed in our protein list. Note that contaminants that are excluded from results lists do not contain chaperones but rather typical proteomics contaminants such as keratins or proteinases. We therefore expect this particular protein band to represent one or more specific vesicle proteins. Proteins with similar molecular weight are, for instance, SV2A (82 kDa, a major component) or V-ATPase subunit A (66 kDa).

- Figure 4C: The authors mention that “Intra-molecular cross-links that also support oligomerisation of Synaptophysin are shown as dashed lines.” The dashed lines show inter-molecular cross-links. Please clarify or change wording.

We agree that this was misleading. We have changed the text accordingly. It now reads “Cross-links that support oligomerisation of Synaptophysin are shown as dashed lines”. In

addition, we added a short explanation to the main text (page 13).

- At the end of page 14 the authors state: “We next engaged Synaptobrevin-2 in SNARE complex formation and thereby locked the protein in a functionally active form.” To which degree are the SNARE motifs assembled in this complex? Does the complex indeed reflect an activated (prefusion) stage? The authors may consider to change the wording.

We agree that this expression was again misleading. Our methods do not prove whether SNARE motifs are assembled. Indeed, the assembled complexes rather represent postfusion states. We have changed the text accordingly. It now reads “We next assembled Synaptobrevin-2 in the SNARE complex and thereby locked the protein in a post-fusion state.”.

Reviewer #3 (Remarks to the Author):

The authors performed a proteomics and crosslinking mass spectrometry study on synaptic vesicles. While I can comment on the quality of the crosslinking mass spectrometry (XL-MS) findings and how to interpret them I cannot comment totally on the biological relevance and novelties of these results. The way the results are currently presented however, I fear their usefulness may be limited. Several replicated preps of the vesicles are performed and proteomics confirms them to resemble previous work confirming a good prep. Negative stain EM also confirms that these vesicles seem intact. The paper begins with a proteomics study of vesicles and while iBAQ is not absolute quantitation the estimation of relative abundances is enough for the presented study. I find this to be very thorough.

The XL-MS study was also competently done. It is stated that crosslinks were validated at 5% FDR using Plink but then a manual validation was performed. Why? How many crosslinks were removed manually?

Our analyses were performed to derive biologically relevant interactions and interaction networks, which we aimed to target in subsequent experiments. We therefore only wanted to include high-confidence cross-links in our protein networks validated the list of potential cross-links manually. As referenced in our manuscript, this procedure was previously recommended (Iacobucci, Sinz (2017) To Be or Not to Be? Five Guidelines to Avoid Misassignments in Cross-Linking/Mass Spectrometry. *Anal Chem* 89, 15, 7832–7835).

My major concerns around information that can be gleaned from this study are as follows:

- The description of the crosslinked interaction network begins with V-type ATPase. I find the cartoon representation of the structure to be unsatisfactory. There has recently been a structure of a mammalian V-type ATPase solved and a homology model should be produced based on this and the crosslinks mapped on. This will vastly improve any structure-based arguments about the populations of V1-V0 and V0 in the sample. All of the structure-based arguments are weak until this is done. This should identify a cluster of unsatisfied links on the whole V0-V1 complex that can only be explained by a lone V0 structure, perhaps using DisVis if a suitable structure is not known.

As requested by the reviewers, we have used the model of the recently published EM structure of the V-ATPase complex from rat brain (Abbas et al.) to map cross-links identified in our study. We visualised the cross-links and present the results in our revised Figure 2. We also provide cross-linking distances in the additional supplementary table (Table S4 in our revised manuscript). The majority of cross-links identified in the V-ATPase complex shows a distance of ≤ 30 Å. Interestingly, cross-link with distances of > 30 Å are those cross-links which satisfy the inhibited conformation of the V-ATPase. These cross-links were mainly observed between subunits a1 and d1. To further proof specificity of our cross-links, we generated random cross-linking distances between identified cross-linked lysine residues. The cross-linking distances of randomly linked lysine residues are significantly longer and the distribution of distances is wider; again, confirming our cross-linking approach. The crosslinking distances of randomly linked cross-links is shown in Figure S5.

Please also note that the purified V-ATPase complex published recently by Abbas et al. was purified using a detergent and a capture protein associated with the 'head' of the V-ATPase (V1 domain). As a consequence, the high-resolution structure does not contain the full cohort of V-ATPase subunits. In our study, we found the vesicle-embedded V-ATPase to be complete. In addition, as the V-ATPase studied by Abbas et al. was purified through the 'head' of the enzyme, they are unable to study other structural conformations such as the dissociated enzyme. Our cross-linking approach, however, delivers information on both conformations, the V_1V_0 -ATPase as well as the dissociated V_0 -ATPase.

- The authors convincingly demonstrate that the Synaptobrevin-2 crosslinks are difficult to interpret. Synaptobrevin-2 is a flexible protein that is found in relatively high concentration on these vesicles. The very long half-life of BS3 means that it will very often crosslink uncommon interactions of such a protein. It was shown nicely in this work by addition of SNARE, dilution of vesicle surface or cleavage of the Synaptobrevin-2 that many of these detected interactions were such crosslinking-induced artifacts.

We want to clarify that the interactions observed in our initial protein network are not crosslinking artefacts. They rather originate from the flexibility and unstructured characteristics of Synaptobrevin-2. This flexibility is important for its function and the interactions should not be underestimated. Our follow up experiments (BoNT B cleavage, dilution etc.) rather confirm the origin of the interactions and allow us to differentiate between specific and unspecific binding.

However, as many of the interactions identified here have already been shown in some form by other previous studies it would have been wise if the authors had used the structural information present in the crosslinking data to draw some structural conclusions. One example would be, if the interactions between the V_0 region of the ATPase and Synaptobrevin-2 is specific there should be a binding 'site' that could be suggested by the crosslinks.

It is true that the interactions of synaptic vesicle proteins have been the focus of many studies, however, the outcomes of these studies are contradictory. In addition, these studies were mostly performed in the 1980's and 1990's and the techniques available at that time were rather limited. As a consequence, there are almost no high-resolution structures of synaptic vesicle proteins. Previous studies showed, that the interactions were strongly dependent on the detergents used in these studies. In our study, we did not use any detergents and we therefore provide an unbiased view on synaptic vesicle interactions. The specific interaction mentioned by the reviewers (Synaptobrevin-2 vs. V-ATPase subunit a) suggests that indeed the flexible domain of V-ATPase subunit a is responsible for this interactions (see Fig. 2 and Table S3). This is in agreement with our other findings that this flexible protein takes regulatory roles in the synaptic vesicle cycle. We expect this specific interaction as well as many other interactions to serve as a basis for many future studies by ourselves and the synaptic community.

Of note, as cross-linking only delivers information on binary protein interactions, protein interaction networks do not necessarily represent defined protein complexes and might include several protein complexes with overlapping interactions. This is particularly true for the protein complexes in synaptic vesicles as they have to assemble and disassemble during the vesicle cycle. This should be kept in mind when modelling protein interactions. In this study, we were able to partially unravel these interactions; however, this will be a main goal of future studies (i.e. identify protein microdomains/protein complexes in these networks).

Other points.

Figure S4 shows spectra. My concern with this panel is that it is potentially simply two gas phase associated peptides instead of a real crosslinked di-peptide. The lack of b-ion fragment series concerns me. To check this, interrogate the spectra with one each of the peptides instead containing

a monolink (hydrolysed crosslinker at the lysine) and see if this allows matching of b-ions to the two linear peptides.

We disagree. Several b-ions were clearly observed and labelled in the respective mass spectra (Figure S4a: B2; Figure S4b: b2, b3, b4, B2, B3; Figure S4c: b2/B2, b3/B3 and b4/B4). These ions clearly identify the location of the cross-linking site and also confirm the presence of a cross-linked peptide. Even though only 1 b-ions was observed in the mass spectrum shown in Figure S4a, the cross-linking site could clearly be assigned. Note that gas-phase associated peptides, as described in Giese et al (2019) Anal Chem, have been observed only for co-eluting peptides. This problem has been described specifically when cross-linked proteins are extracted from gel bands without enrichment of cross-linked peptides. In our study, proteins were cross-linked in solution and, after hydrolysis with trypsin, low-abundant cross-links were enriched through gel filtration. Linear peptides are separated from crosslinked peptides in this enrichment step and it is most unlikely that they are present in the same fraction obtained from gel filtration. They can therefore not co-elute during LC-MS analysis. In addition, gas phase association has been described to be particularly critical for peptides with overlapping sequences (multimers/oligomers) (Giese et al). In our study, we confirmed oligomerisation/multimerisation for Synaptobrevin-2 and Synaptophysin by analysing cross-linked proteins using specific antibodies after Western Blotting. In addition to the separation of linear and cross-lined peptides during gel filtration, we therefore provide a second, independent proof for these interactions.

Figure S5, why were vesicles crosslinked with 15mM BS3 for Western blot when the XL-MS was performed on 10mM crosslinked vesicles?

All cross-linking-MS experiments were performed with 10 mM BS3. In addition, we validated identified cross-links of Synaptobrevin-2 and Synaptophysin using specific antibodies against the two proteins. These experiments confirmed oligomerisation/dimerization of the two proteins. In addition, we observed an interaction between the two proteins, however, this interaction was difficult to visualise by western blotting. We therefore performed one experiment with higher protein and BS3 concentrations to make the band detectable by both antibodies. This experiment was only confirmatory.

The meaning of the following sentence is unclear and should be clarified, "We therefore conclude that the main interactors of synaptic vesicles are Synaptobrevin-2, Synaptophysin/Synaptoporin, SV2A (SV2B) and the V-ATPase complex."

Even though synaptic vesicles contain a multitude of proteins, the majority of cross-links was identified in a small, defined set of proteins. Interactions that were maintained after dilution of the proteins in a larger membrane were formed between Synaptobrevin-2, Synaptophysin/Synaptoporin, SV2A (SV2B) and the V-ATPase complex. We have changed this sentence accordingly.

Reviewer #1 (Remarks to the Author):

In this revised version of the manuscript, the authors have addressed all the issues raised. I have no additional requests or concerns.

Reviewed by Alexander Leitner, ETH Zurich, Switzerland

Reviewer #3 (Remarks to the Author):

I have to admit that I am underwhelmed by the response of the authors. I am not arguing about the correctness of the cross-linking data as such. In this manuscript, there is no technical advancement, fair enough. But as it stands, there is also no biological advancement either. As I wrote “However, as many of the interactions identified here have already been shown in some form by other previous studies it would have been wise if the authors had used the structural information present in the crosslinking data to draw some structural conclusions.” I still feel strongly about this point. A lot of the cross-links may arise simply because the proteins are held in close proximity and at high local concentration by being in a vesicle membrane together and cross-linked with a long-lived reagent. Even once this is addressed, biological insight derives not from a structural model alone (of which there is none currently in the manuscript) but from testing the hypotheses that build on the model.

I wonder if the descriptive dataset alone suffices to appear in Nature Comms. I leave this down to the editor’s judgement.